# Modulation of MRSA virulence gene expression by the wall teichoic acid enzyme TarO

Yunfu Lu[1,2,3,8], Feifei Chen [1,2,4,8], Qingmin Zhao[1,2,3], Qiao Cao[1,2,4], Rongrong Chen[1,2,3], Huiwen Pan[1,2,3], Yanhui Wang[1,2,3], Haixin Huang[2], Ruimin Huang [2,3], Qian Liu[5], Min Li[5], Taeok Bae[6], Haihua Liang [4,7] ✉ & Lefu Lan [1,2,3,4] ✉

Phenol-soluble modulins (PSMs) and Staphylococcal protein A (SpA) are key virulence determinants for community-associated methicillin-resistant *Staphylococcus aureus* (CA-MRSA), an important human pathogen that causes a wide range of diseases. Here, using chemical and genetic approaches, we show that inhibition of TarO, the first enzyme in the wall teichoic acid (WTA) biosynthetic pathway, decreases the expression of genes encoding PSMs and SpA in the prototypical CA-MRSA strain USA300 LAC. Mechanistically, these effects are linked to the activation of VraRS two-component system that directly represses the expression of accessory gene regulator (*agr*) locus and *spa*. The activation of VraRS was due in part to the loss of the functional integrity of penicillin-binding protein 2 (PBP2) in a PBP2a-dependent manner. TarO inhibition can also activate VraRS in a manner independent of PBP2a. We provide multiple lines of evidence that accumulation of lipid-linked peptidoglycan precursors is a trigger for the activation of VraRS. In sum, our results reveal that WTA biosynthesis plays an important role in the regulation of virulence gene expression in CA-MRSA, underlining TarO as an attractive target for anti-virulence therapy. Our data also suggest that acquisition of PBP2a-encoding *mecA* gene can impart an additional regulatory layer for the modulation of key signaling pathways in *S. aureus*.

Methicillin-resistant *Staphylococcus aureus* (MRSA) is resistant to most clinically used β-lactam antibiotics[1,2]. The β-lactam resistance of MRSA is due to the acquisition of a *mecA* gene that encodes a non-native penicillin-binding protein (PBP) designated PBP2a[1]. This protein exhibits low affinity for β-lactams and thus enables MRSA to sustain cell wall peptidoglycan (PG) synthesis when the transpeptidases (TPase) activity of the other native *S. aureus* PBPs, PBP1 to PBP4, are inhibited by β-lactams including methicillin[1–3]. PBP2a-mediated β-lactam resistance is additionally modulated by the activity of pathways involved in the regulation or biosynthesis of bacterial cell wall polymers, including PG, wall teichoic acid (WTA), and lipoteichoic acid (LTA)[1,3]. Currently, treatment of MRSA infections is challenging, and a deep understanding of the pathogenic success of MRSA is vital for the development of new anti-MRSA therapeutics[3–5].

MRSA was first reported in the early 1960s and was mainly confined to healthcare-associated facilities (HA-MRSA); however, this

[1]School of Pharmaceutical Science and Technology, Hangzhou Institute for Advanced Study, University of Chinese Academy of Sciences, Hangzhou 310024, China. [2]State Key Laboratory of Drug Research, Shanghai Institute of Materia Medica, Chinese Academy of Sciences, Shanghai 201203, China. [3]University of Chinese Academy of Sciences, No. 19A Yuquan Road, Beijing 100049, China. [4]College of Life Science, Northwest University, Xi'an 710127, China. [5]Department of Laboratory Medicine, School of Medicine, Renji Hospital, Shanghai Jiao Tong University, Shanghai 200127, China. [6]Department of Microbiology and Immunology, Indiana University School of Medicine-Northwest, Gary, IN 46408, USA. [7]School of Medicine, Southern University of Science and Technology, Shenzhen 518055, China. [8]These authors contributed equally: Yunfu Lu, Feifei Chen. ✉e-mail: lianghh@sustech.edu.cn; llan@ucas.ac.cn

bacterium has evolved dramatically over the last 3 decades, and the community-associated MRSA (CA-MRSA) has emerged as an important cause of infections and now become a public health concern[3]. Unlike most strains of HA-MRSA, CA-MRSA strains generally display low-level resistance to β-lactam antibiotics and exhibit higher virulence and often cause disease in otherwise healthy individuals[3,5–7]. The molecular basis for the enhanced virulence of CA-MRSA strains remains an active area of research, and studies have shown that increased expression of core genome-encoded toxins such as phenol-soluble modulins (PSMs) is causally associated with the enhanced virulence potential of CA-MRSA, particularly USA300, a prevailing clone in North America[6,7]. Recently, a study showed that the production of WTA, a major PG-anchored cell wall glycopolymer, is also involved in the enhanced virulence of CA-MRSA *via* a WTA-dependent and T-cell-mediated mechanism[8]. To date, the increased virulence of CA-MRSA strains is incompletely understood, while it is generally believed that the success of CA-MRSA strains was due to the combination of methicillin resistance at low fitness cost and high virulence as compared to the traditional HA-MRSA[3,6,7].

PSMs, a family of cytolytic peptide toxins, are key virulence determinants in CA-MRSA[9]. *S. aureus* produces several core-genome-encoded PSMs, mainly α- and β-type PSMs and the δ-toxin (Hld)[10]. Among them, PSMαs are encoded by the *psmα* operon, PSMβs are encoded by the *psmβ* operon, while the δ-toxin is encoded within RNAIII that acts as an effector molecule of the staphylococcal accessory gene regulator (*agr*) locus[10,11]. In *S. aureus*, the *agr* locus produces two divergent transcripts, RNAII and RNAIII[10,11]. RNAII encodes four genes (*agrA*, *agrB*, *agrC*, and *agrD*), and the gene products assemble a two-component system (TCS) based quorum sensing (QS) system, where AgrC is the sensor histidine kinase, and AgrA is the response regulator[11]. AgrA activates the synthesis of its own operon (RNAII) and of RNAIII as well as the *psmα* and *psmβ* operons[11,12]. Indeed, the overproduction of PSMs in CA-MRSA clones could be explained by a hyperactive *agr* QS system[7]. Importantly, deletion of either *agr* locus or *psmα* operon in the epidemic U.S. strains (i.e., USA300 and USA400) dramatically reduces bacterial virulence in different animal models of infection, which points to the value of inhibiting PSMs production for drug development to control the severity of CA-MRSA infections[9,10].

Like PSM production, WTA biosynthetic pathway has also drawn significant interest as a drug target against MRSA infections, particularly as the lack of WTA destabilizes the cooperative action of PBPs and re-sensitizes MRSA to β-lactams[5,13,14]. Although considerable progress has been made regarding the important function of WTA in Staphylococcal infections[15–17], its role in virulence gene expression is unknown. Here, using a high-throughput chemical screening and genetic approaches, we reveal an important, unanticipated role of WTA biosynthetic pathway in modulating the expression of key virulence genes, including those encoding PSMs, in CA-MRSA strains.

## Results

### Tunicamycin inhibits *agr* activity and the expression of *spa* in USA300

To identify inhibitors for PSM genes expression, we constructed and integrated a *psmα* promoter-*lacZ* reporter gene (i.e., *psmα-lacZ*) into the chromosome of *S. aureus* strain USA300 LAC[3]. Using a disk diffusion assay, we examined the ability of 3,987 compounds (i.e., 2,395 FDA-approved drugs, 1199 clinically tested drugs, and 393 bioactive agents) to inhibit *psmα-lacZ* expression (Sheet 1 in Supplementary Data 1), and we identified 19 potential inhibitors for the expression of *psmα-lacZ* (Sheet 2 in Supplementary Data 1). Among them, 16 compounds (i.e., tunicamycin and 15 β-lactam antibiotics) were linked to the inhibition of peptidoglycan (PG) biosynthesis (Fig. 1a, Sheet 2 in Supplementary Data 1). We focused on the mode of action of tunicamycin in this study because this compound also can re-sensitize MRSA strains to β-lactam antibiotics[13].

To validate the reduced expression of *psmα* operon conferred by tunicamycin, we performed quantitative reverse transcription PCR (qRT-PCR) analysis to examine the expression of *psmα3* that encodes the most cytolytic *S. aureus* PSM[9]. In the concentration range of 0.02–0.5 μg/ml, tunicamycin showed concentration-dependent inhibition of *psmα3* expression in the USA300 LAC cells (Fig. 1b). Tunicamycin also can dramatically attenuate (>80% decrease) the expression of *psmβ1* and RNAIII, even at low concentrations (i.e., 0.1 μg/ml) (Fig. 1c, d). These data suggest an inhibitory role of tunicamycin on *agr* QS activity because the expression of PSM genes is positively and strictly controlled by the *agr*[9,10,12].

To verify the role of tunicamycin in the activity of *agr*, we constructed a *lux*-based bioluminescence system in USA300 LAC by expressing *lux* genes under the control of *psmα* promoter. The *lux* reporter strain (i.e., USA300::*psmα-lux*) was treated with tunicamycin (0.5 μg/ml), either in the absence or in the presence of exogenous addition of *agr* autoinducing peptide (AIP). Our results showed that tunicamycin could reduce the luminescence under these test conditions (Fig. 1e–g), further supporting that tunicamycin is an inhibitor of the *agr* activity in USA300 LAC.

Unexpectedly, however, the tunicamycin treatment of USA300 LAC cells decreased the transcription of the gene encoding staphylococcal protein A (SpA) (Fig. 1h), an important virulence factor whose production is negatively controlled by *agr*[11,12]. Using western blot experiments, we detected a significant reduction of SpA production in USA300 LAC upon treatment with tunicamycin in a concentration-dependent manner (Fig. 1i and Fig. S1a).

### WTA biosynthesis is intimately linked to the expression of PSM genes and SpA

In *S. aureus*, tunicamycin inhibits TarO and MraY that catalyze an early stage reaction in WTA and PG cell wall assembly[1,18]. Among them, TarO, which transfers GlcNAc from UDP-GlcNAc to the lipid carrier undecaprenyl-phosphate (Und-P), is the first enzyme in the WTA biosynthetic pathway and is readily dispensable in *S. aureus*, while MraY, which transfers the phospho-N-acetylmuramoyl-pentapeptide from UDP-MurNAc-pentapeptide to the Und-P to form bacterial PG precursor lipid I, is essential for bacterial cell growth[1,18] (Fig. 1a). Because WTA-inhibitory concentrations of tunicamycin (Fig. S1b and c) reduced the expressions of PSM genes and SpA (Fig. 1), and such concentrations of tunicamycin do not affect the exponential growth rates of USA300 LAC strain (Fig. S1d), we reasoned that inhibition of TarO, but not MraY, by tunicamycin, would mediate the decreased expression of *psmα* operon and SpA. To test this hypothesis, we generated a USA300 LAC mutant (i.e., Δ*tarO*) in which the gene encoding TarO was deleted. Using qRT-PCR and western blot experiments, we found that *tarO* deletion causes a dramatic decrease in the expression level of both *psmα*3 (Fig. 2a) and SpA (Fig. 2b, c). Importantly, although tunicamycin significantly reduced the expression level of either *psmα*3 or SpA in wild-type (WT) USA300 LAC strain and in the complemented strain of Δ*tarO* mutant (i.e., Δ*tarO*/p-*tarO*), it failed to do so in the Δ*tarO* mutant (Fig. 2a–c). Thus, the inhibitory effect of tunicamycin on *psmα* and SpA expression requires *tarO*.

To genetically verify that reduced amounts of WTA were associated with decreased expression of *psmα* and SpA, we created a *tarO* mutant construct, in which the glycine residue in position 152 of TarO was changed to alanine (G152A). This mutation was previously shown to decrease the production of WTAs[19]. Our results showed that the *tarO* (G152A) mutation (Δ*tarO*/p-*tarO*^G152A strain) reduced the levels of WTA (Fig. S1e and f) and the expression of *psmα3* (Fig. S1g) and SpA (Fig. S1h and i) simultaneously in a Δ*tarO* background.

### TarO is crucial for the expression of virulence genes in USA300

To further examine the roles of WTA biosynthesis in the expression of virulence genes, we performed RNA-seq and compared the

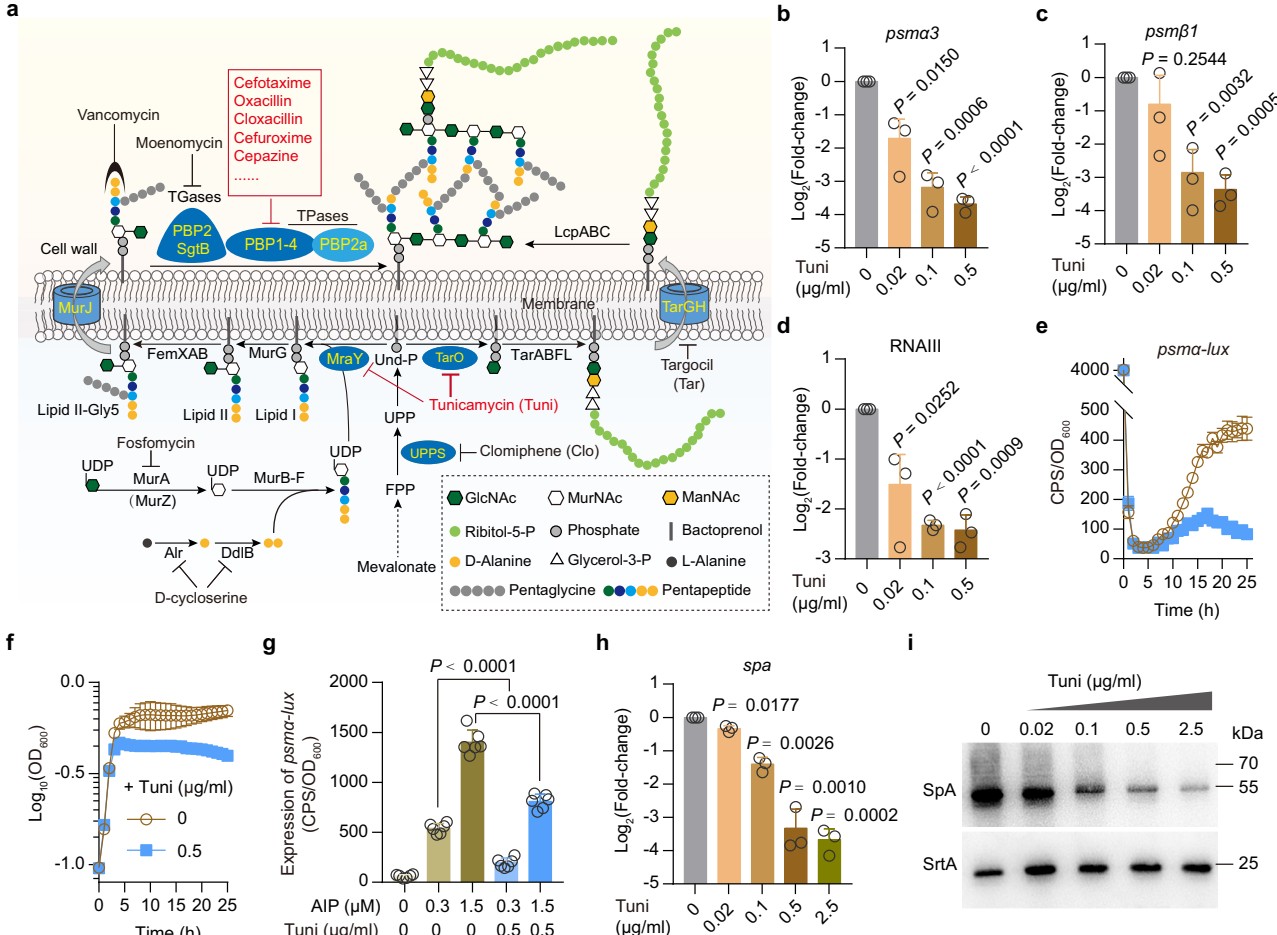

**Fig. 1 | Tunicamycin inhibits *agr* activity and SpA production. a** Schematic of the primary WTA and cell wall biosynthetic pathways and the targets of antibiotics. Potential *psmα-lacZ* inhibitors identified in this study are highlighted in red. **b**–**d** qRT-PCR analysis of *psmα3*, *psmβ1*, and RNAIII transcripts in USA300 LAC grown in TSB medium supplemented with or without indicated concentrations of tunicamycin (Tuni) for 3 h, presented as relative expression levels (log2 fold changes) compared with the tunicamycin-untreated WT USA300. Data represent mean ± SD from three independent experiments. Statistical analysis was performed using two-tailed one-sample *t*-test with the tunicamycin-untreated WT USA300 set to a fold change of 1. **e**, **f** Expression of *psmα-lux* (**e**) and the corresponding growth curve (**f**) of WT USA300 LAC strain cultured in TSB medium supplemented with or without tunicamycin (Tuni). The value of CPS/OD$_{600}$ (counts per second/an optical density at 600 nm) became an indicator of the promoter activity. Data from n = 6 biological replicates are reported as the mean ± SD. **g** Expression of *psmα-lux* in WT

USA300 LAC strain cultured in TSB medium supplemented with or without indicated final concentrations of compounds for 2 h. AIP, auto-inducing peptide. Data from n = 6 biological replicates are reported as the mean ± SD. Statistical analysis was performed using Student's two-tailed unpaired *t*-test. **h** qRT-PCR analysis of *spa* transcripts in USA300 LAC grown in TSB medium supplemented with indicated concentrations of tunicamycin for 1.5 h, presented as relative gene expression levels (log2 fold changes) compared with the untreated WT USA300 control. Data represent mean ± SD from three independent experiments. Statistical analysis was performed using two-tailed one-sample *t*-test with the untreated WT USA300 control set to a fold change of 1). **i** Representative images of Western blotting for SpA in USA300 LAC grown in TSB medium supplemented with indicated concentration of tunicamycin (Tuni) for 3 h. Sortase A (SrtA) is a loading control. The experiment was repeated independently three times with similar results (Fig. S1a).

transcriptome of Δ*tarO* mutant with that of its parent WT USA300 LAC strain in tryptic soy broth (TSB) medium at 1.5 h and 3 h incubation, respectively. A total 99 genes showed significantly differential transcript abundance at 1.5 h after inoculation while a total of 921 genes at 3 h (≥2-fold change and *q*-value ≤ 0.05) (Fig. 2d, Supplementary Data 2). At both time points, overlap on gene expression was observed, but large numbers of genes were uniquely up- or downregulated upon *tarO* deletion at one of the time points (Fig. 2d), indicating that the effects of WTA on *S. aureus* transcriptome largely depend on bacterial growth condition.

At 1.5 h incubation, *tarO* deletion downregulated 42 genes and upregulated 57 genes (Fig. 2d, Supplementary Data 2). Noticeably, the gene encoding SpA was downregulated the most (14.7-fold) (Supplementary Data 2). A number of virulence genes, including those encoding MSCRAMM family proteins (i.e., *sdrD*, *sdrC*, *sasD*, *harA*, and *sasA*) and components of the type VII secretion system (T7SS) (i.e.,

*esxA*, *esaA*, *essB*, *essC*, *esxB*, *essE*, *exsD*, and *essD*), were also significantly downregulated (Supplementary Data 2). By contrast, the most up-regulated (145-fold) gene upon *tarO* deletion was *vraX* (Supplementary Data 2), which encodes a secreted protein that specifically inhibits the classical pathway of the complement system by binding to C1q[20]. It also has been reported that *vraX* is a very sensitive reporter of cell wall stress and whose induction by cell wall-active agents is dependent on the VraRS TCS[21]. In addition to *vraX*, a number of VraRS-activated genes (e.g., *vraR*, *sgtB*, *cwrA*, *psrA*, *lytR*, *nirB*, *cobA*, *narH*, *narJ*, *narI*, *narK*, *HUW68_RS12685*, *HUW68_RS08595*, *HUW68_RS12300*, and *HUW68_RS14260*)[22,23] were also significantly up-regulated (2- to 17.9- fold) (Supplementary Data 2), implying an activation of VraRS in the Δ*tarO* mutant. Moreover, the activation of VraRS can explain, at least in part, the decreased expression of several VraRS-repressed genes (e.g., *spa*, *exsA*, *esaA*, *tarM*, *HUW68_RS01445*, *HUW68_RS01460*, and *HUW68_RS13315*) in the Δ*tarO* mutant when compared to the WT

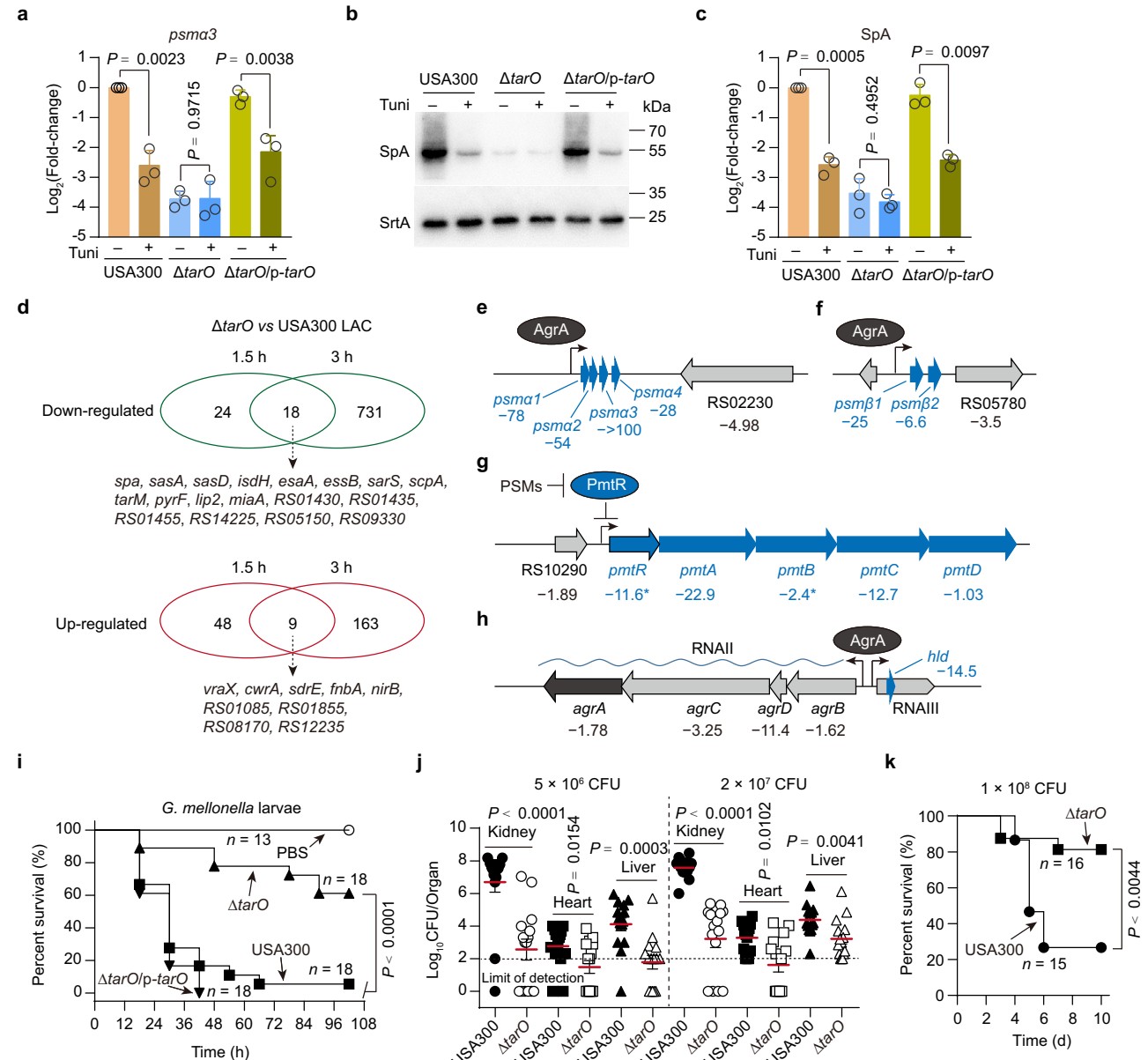

**Fig. 2 | Effect of *tarO* deletion on virulence gene expression and on the virulence potential of USA300. a** Relative *psmα3* transcripts in *S. aureus* strains grown in TSB medium supplemented with (+) or without (−) tunicamycin (Tuni, 0.5 μg/ml) for 3 h, presented as relative expression levels (log2 fold changes) compared with the tunicamycin-untreated WT USA300. Data represent mean ± SD from three independent experiments. **b, c** Representative images (**b**) and quantitative analysis (**c**) of Western blotting for SpA in *S. aureus* strains grown in TSB medium supplemented with (+) or without (−) tunicamycin (0.5 μg/ml) for 3 h. Band intensity of SpA was normalized to the intensities obtained with loading control SrtA, and the results are reported as relative expression levels (log2 fold changes) compared with the tunicamycin-untreated WT USA300. Data represent mean ± SD from n = 3 independent experiments. **d** The overlap of *tarO*-activated genes (upper panel) and *tarO*-repressed genes (lower panel) between the 1.5- and 3-h time points. **e–h** Genes (or operons) modulated by *tarO* at the 3-h time point. The minus symbol indicates downregulation upon *tarO* deletion, the numbers indicate the fold change relative to the WT USA300 LAC control. **i** Survival rates of *G. mellonella* larvae upon infection by *S. aureus* strains. **j** Virulence of USA300 and Δ*tarO* strains. The female BALB/c mice were challenged with 5 × 10⁶ CFU (n = 15 for USA300 and Δ*tarO* strains) or 2 × 10⁷ CFU (n = 14 for USA300, n = 15 for Δ*tarO*). Horizontal bars (in red) indicate observation means. **k** Survival rates of female BALB/c mice upon infection by WT USA300 LAC strain and Δ*tarO* mutant. Statistical analysis was performed using two-tailed one-sample *t*-test (when compared with the tunicamycin-untreated WT USA300 control, which was set to a fold change of 1) or otherwise with Student's two-tailed unpaired *t*-test (in **a** and **c**), log-rank test (in **i** and **k**), and Mann–Whitney two-tailed test (in **j**). In (**a, b, c**, and **i**), USA300 and Δ*tarO* strains harbor an empty pYJ335-1 vector as control.

USA300 LAC strain (Supplementary Data 2)[23]. Several autolysin-encoding genes (e.g., *ssaA*, *ssaA1*, *isaA*, *sle1*, *lytM*, *aaa*, and *sceD*, and *HUW68_RS1347S*)[24] were also up-regulated upon *tarO* deletion (Supplementary Data 2).

At 3-h time incubation, 749 genes were downregulated while 172 genes were up-regulated by the deletion of *tarO* (Fig. 2d, Supplementary Data 2). These differentially expressed genes (DEGs) represent ~32% of the USA300 LAC genes and were involved in diverse biological processes, including in vivo virulence and cell wall-stress response (Supplementary Data 2). Among the top 10 downregulated genes (>20-fold), we found five PSM-encoding genes (PSMα1-4 and PSMβ1) (Fig. 2e, f) and *pmtA* (Fig. 2g) that codes for the ATPases PmtA of the PSM-exporting system PmtABCD[10]. Interestingly, genes encoding PSMβ2 (Fig. 2f), δ-toxin (encoded in the RNAIII) (Fig. 2h), and the ATPases PmtC of the PmtABCD system (Fig. 2g) were also strongly downregulated (6.6–14.5 fold) upon *tarO* deletion (Supplementary

Data 2). Meanwhile, we observed that *agrD* within the *agrBDCA* operon was significantly downregulated (>8-fold) and other genes of this operon, *agrA*, *agrB*, and *agrC* were moderately downregulated by *tarO* deletion (Fig. 2h). The downregulation of *agr* locus may explain the decreased expression of PSM genes and those involved in their secretion, that is, the PmtABCD system[10,25].

In addition to the *agr* locus (Fig. 2h), genes encoding well-known virulence regulators, such as SaeRS, SarA, SarR, and SarS, were also significantly downregulated (> 5-fold) upon *tarO* deletion at a 3-h time point (Supplementary Data 2). The downregulation of *saeR* (5.9-fold decrease; −5.9 in brief) and *saeS* (−8.7) can explain the decreased expression of a set of virulence genes including *ssl1* (−15.8), *eap* (−5), *efb* (−3.4), *scn* (−9.7), *lukH* (−3), *lukG* (−5.5), *sbi* (−2.3), *hlgA* (−2.2), *hlgB* (−5.2), *hlgC* (−3.8)[26]. Also, the downregulation of *sarS* (−7) may partly explain the decreased expression of *spa* (−3.9) (Supplementary Data 2)[27]. In contrast, the transcription of some VraRS-activated genes (e.g., *vraX* and *cwrA*)[22,23] and tRNA genes was dramatically increased (>8-fold) upon *tarO* deletion after a 3-h inoculation (Supplementary Data 2).

To validate the results of RNA-Seq, we cultured WT USA300 LAC, Δ*tarO* mutant, and its complementation strain (Δ*tarO*/p-*tarO*) in TSB medium (Fig. S2a and b) and carried out qRT-PCR analysis for seven key DEGs (i.e., *spa*, *psmα3*, RNAIII, *vraX*, *psmβ1*, *saeR*, and *saeS*). In general, our qRT-PCR data were consistent with the RNA-seq results while discrepancies were also found (Fig. S2c–g). For instance, although our RNA-seq experiments showed that *tarO* deletion had no significant impact on the expression of *psmα3* and RNAIII at the 1.5-h time point (Supplementary Data 2), the qRT–PCR analysis indicated that both *psmα3* and RNAIII were downregulated upon *tarO* deletion at the time point (Fig. S2d and e). These discrepancies may be due to the low expression of *psmα3* and RNAIII (Supplementary Data 2), requiring the sensitivity and precision of qRT-PCR for detection. Importantly, the introduction of the WT *tarO* carried on a plasmid into the Δ*tarO* mutant restored the expression of these genes to almost normal levels (Fig. S2c–g). In addition, we observed that synthetic AIP is able to rescue the *tarO* deletion-induced downregulation of *psmβ1* and *pmtA* (Fig. S2h). A similar result was also obtained when the expression levels of *psmα3* were examined, although the increase was not statistically significant (Fig. S2h). Based on these data, we concluded that in the absence of *tarO*, USA300 LAC maintains cell wall integrity by up-regulating the VraRS and reduces the expression of virulence genes mainly by down-regulating *agr* QS system.

## TarO is required for the virulence of USA300 in animal infection models

Deletion of *tarO* significantly decreased the expression of numerous virulence-associated genes (Fig. 2a–h, Supplementary Data 2) as well as the hemolytic activity (Fig. S2i) that has been identified as a key virulence factor in *S. aureus*[4]. Therefore, we next examined the role of *tarO* in the virulence potential of USA300. In a *Galleria mellonella* infection model, the Δ*tarO* mutant showed reduced virulence compared with the WT USA300 LAC or the Δ*tarO*-complemented strain (i.e., Δ*tarO*/p-*tarO*) (Fig. 2i).

To further examine the role of TarO in the virulence of USA300 LAC, mice were challenged with $5 \times 10^6$ CFU of WT USA300 LAC or Δ*tarO* mutant *via* retro-orbital injection, then measured bacterial loads in host organs five days after infection. Δ*tarO* displayed markedly reduced bacterial loads in kidneys, hearts, and livers compared to the WT USA300 LAC (Fig. 2j). Specifically, the bacterial loads of Δ*tarO* strain in kidneys were four-log lower than that of USA300 WT strain (Fig. 2j). Similar results were obtained when the mice were challenged with $2 \times 10^7$ CFU of the testing strains (Fig. 2j).

With an infection dose $1 \times 10^8$ CFU, we observed that WT USA300 LAC caused 73.3% (11/15) mortality of the infected mice within ten days, whereas only 18.8% (3/16) of the Δ*tarO*-infected mice died (Fig. 2k).

Collectively, these results suggest that TarO plays a key role in the virulence of *S. aureus* USA300 LAC.

## TarO inhibition-induced changes in gene expression largely depend on VraR

Our RNA-seq data suggest that *tarO* deletion induces VraRS activation (Supplementary Data 2). To verify whether VraRS has a role in mediating virulence gene expression in response to WTA depletion, we generated a USA300 LAC mutant (i.e., Δ*vraR*) in which the gene encoding the response regulator VraR was deleted. Using transcriptional fusion assays, we showed that *vraR* deletion abolishes the tunicamycininduction of *vraX* promoter activity (Fig. 3a, b), indicating that TarO inhibition does cause VraRS activation.

Likewise, using *psmα-lux* transcriptional fusion assays, we found that tunicamycin (0.1 μg/ml) treatment reduces the expression of *psmα-lux* in USA300 LAC strain by ~20-fold (at 25 h) while it inhibited the expression of *psmα-lux* moderately (2.5-fold) in the Δ*vraR* mutant (Fig. 3c, d). These observations suggest that the inhibitory effect of tunicamycin on the promoter activity of *psmα3*, at least in part, depends on *vraR*. A similar result was obtained when the expression of *psmα3* was examined by qRT-PCR experiments (Fig. S3a).

Tunicamycin (0.5 μg/ml) treatment significantly reduced the production of SpA in WT USA300 LAC strain and the complemented strain of Δ*vraR* mutant (i.e., Δ*vraR*/c) but failed to do so in the Δ*vraR* mutant (Fig. 3e, f), indicating that VraR strictly controls the production of SpA under the test conditions employed. To further verify the inhibitory effect of VraR on the SpA production, we generated a *tarO* deletion in the Δ*vraR* strain. The resulting mutant, termed Δ*tarO*Δ*vraR*, exhibited defects in growth in early log phase (Fig. S3b) while it displayed wild-type levels of SpA (Fig. S3c and d). Ectopic expression of the *vraR* gene in strain Δ*tarO*Δ*vraR* resulted in reduced levels of SpA as observed for Δ*tarO* (Fig. S3c and d). These observations suggest that the reduced level of SpA in the Δ*tarO* mutant is due to the activation of VraR.

To further examine the role of VraR in mediating the effects of deletion of *tarO*, we performed qRT-PCR analyses of *spa*, *psmα*, and RNAIII for WT USA300 LAC strain and its derivative strains (e.g., Δ*tarO*, Δ*tarO*Δ*vraR*, and Δ*tarO*Δ*vraR* complemented with plasmid-borne *vraR*) grown in TSB medium for 3 h. In this assay, RNA from Δ*tarO*Δ*vraR* mutant cells of prolonged culture (i.e., at 4 h and 5 h after incubation) were also included because this mutant exhibited defects in growth and because *agr* system regulates *spa*, *psmα*, and RNAIII by responding to cell population density[11,12]. When grown in TSB medium for 5 h, the Δ*tarO*Δ*vraR* mutant reached a population density ($OD_{600} \approx 2.4$) comparable to the population density of other tested strains cultured for 3 h and exhibited higher expression level of *spa*, *psmα*, and RNAIII than the Δ*tarO* mutant (Fig. S3e–g). Ectopic expression of the *vraR* gene in Δ*tarO*Δ*vraR* mutant resulted in reduced levels of *spa*, *psmα*, and RNAIII as observed for Δ*tarO* mutant (Fig. S3e–g). Thus, VraR inhibits the expression of *spa*, *psmα*, and RNAIII in the Δ*tarO* mutant.

It has been reported that VraR binds to a 15-base sequence (containing an imperfect inverted repeat, AtTTAACaGTTAAGT) of the *agr* promoter and inhibits the function of *agr* in vancomycin-intermediate *S. aureus* (VISA)[28]. To determine whether VraR directly binds to the promoter regions of *vraX* and *spa* whose expressions appear to be strictly controlled by *vraR*, we performed electrophoretic mobility shift assays (EMSAs) with recombinant VraR protein (His₆-VraR). We found that VraR binds to the promoters of *vraX* and *spa*, respectively (Fig. 3g). DNase I footprinting experiments showed that VraR protected *vraX* promoter from DNase I digestion at multiple sites located within 58-134 bp upstream of the start codon (Fig. 3h). We also noted that VraR protects a region of the *spa* promoter against DNase I digestion, and the protected region contains a palindromic sequence of DNA (i.e., ATTTCcGAAAT) (Fig. 3i). Taken together, these data

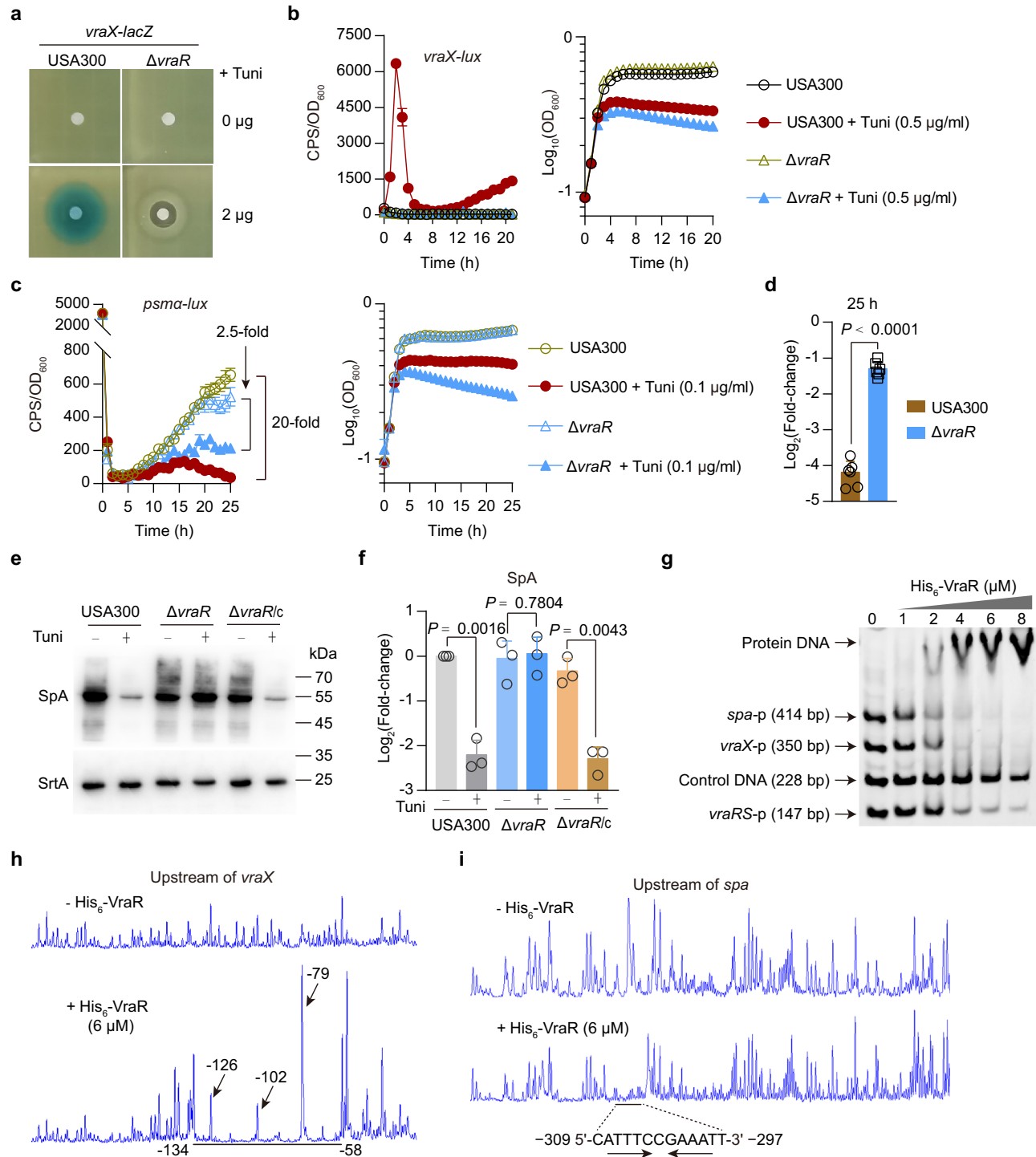

**Fig. 3 | VraR is a negative regulator of *agr* and *spa* in USA300. a** Disk diffusion assay for induction of *vraX-lacZ* by tunicamycin. **b** Expression of *vraX-lux* in *S. aureus* strains grown in TSB medium supplemented with or without tunicamycin (Tuni). Data from *n* = 4 biological replicates are reported as the mean ± SD. **c** Expression of *psmα-lux* (left panel) and the corresponding growth curve (right panel) in *S. aureus* strains grown in TSB medium supplemented with or without tunicamycin (Tuni). Data from *n* = 6 biological replicates are reported as the mean ± SD. **d** Effect of tunicamycin treatment on the expression levels (log2 fold changes) of *psmα-lux* in *S. aureus* strains cultured in TSB medium for 25 h (in **c**). Data from *n* = 6 biological replicates are reported as the mean ± SD. Statistical analysis was performed using Student's two-tailed unpaired *t*-test. **e, f** Representative images (**e**) and quantitative analysis (**f**) of Western blotting for SpA in *S. aureus* strains grown in TSB medium supplemented with (+) or without (−) tunicamycin (0.5 µg/ml) for 3 h. SrtA is loading controls; the results are reported as

relative expression levels (log2 fold changes) compared with the tunicamycin-untreated WT USA300. Data represent mean ± SD from *n* = 3 independent experiments. Statistical analysis was performed using two-tailed one-sample *t*-test (when compared with the tunicamycin-untreated WT USA300 control, which was set to a fold change of 1) or otherwise with Student's two-tailed unpaired *t*-test. USA300 and ΔvraR harbor an empty pYJ335-1 vector as control; ΔvraR/c denotes the complemented strain of ΔvraR (ΔvraR/p-*vraR*). **g** Electrophoretic mobility shift assay (EMSA) showing the DNA-binding ability of purified His_6-VraR to the promoter DNA fragments of *spa* (*spa*-p), *vraX* (*vraX*-p), and *vraRS* (*vraRS*-p). Control DNA, a 228-bp DNA fragment (*coa*-p) covering the promoter region of *coa* gene. **h, i** Electropherograms show the protection pattern of *vraX* (**h**) and *spa* (**i**) promoters after digestion with DNase I following incubation in the absence (−) and presence (+) of His_6-VraR (6 µmol/L). The protected regions (relative to the start codon) are underlined.

clearly suggest that VraR plays an important role in mediating changes in gene expression upon TarO inhibition.

## Inactivation of TarO causes delocalization of PBP2

Since WTA biosynthesis is important for the expression of virulence genes in *S. aureus* (Figs. 1–3, Supplementary Data 2), we sought to examine the underlying mechanism. Three major pieces of evidence prompted us to consider the role of PBP2 in this biological process: (1) blocking TarO function with chemical inhibitor or by gene deletion sensitizes MRSA to β-lactams with a high affinity for PBP2 and not to all β-lactams[14], (2) a number of PBP2-targeting β-lactams inhibit the promoter activity of *psmα* (Sheet 2 in Supplementary Data 1), and (3) loss of *tarO* increases the susceptibility of USA300 LAC to moenomycin (Fig. S4a and b) that targets PBP2 but no other known *S. aureus* PBPs[1,29].

To assess the role of WTA biosynthesis in the functionality of PBP2, we examined whether the lack of WTA causes delocalization of PBP2 because the septal localization of PBP2 is required for its normal function[30]. For this purpose, we constructed a reporter encoding an N-terminal green fluorescent protein (GFP) tagged PBP2 (GFP-PBP2). When the GFP-PBP2 fusion was expressed in the WT USA300 LAC (Fig. S5a), it did localize to the division septum in ~57% of the cells ($n = 372$). By comparing the fluorescence content of the septal area of the cell wall to the lateral area (S/L), we found that the proportion of septal fluorescence intensity is significantly higher for untreated cells (S/L = $2.47 \pm 0.05$) than those treated by epicatechin gallate (ECg) (S/L = $1.65 \pm 0.03$) (Fig. S5b), an inducer for the delocalization of PBP2 in *S. aureus*[31]. This result suggests the reliability of our procedure in detecting the localization of PBP2.

When the same fusion was expressed in the Δ*tarO* mutant (Fig. S5a), the S/L ratio for GFP-PBP2 in the Δ*tarO* mutant was calculated as $1.61 \pm 0.03$, a value significantly lower than that recorded for the WT USA300 LAC strain ($2.47 \pm 0.05$) (Fig. S5b), indicating a reduced PBP2 signal at the septum relative to the periphery upon *tarO* deletion. A similar phenomenon was also observed when the USA300 LAC cells were treated with tunicamycin (Fig. S5b). Thus, blockage of WTA biosynthesis causes delocalization of PBP2 in *S. aureus* USA300 LAC.

## ECg-induced delocalization of PBP2 mimics the effects of TarO inhibition

Previous studies have suggested that delocalization of PBP2 from the septum is primarily responsible for ECg-mediated sensitization of MRSA strains to β-lactam antibiotics[31]. In *S. aureus*, all the four native PBPs (PBP1 to PBP4) possess a transpeptidase (TPase) domain, while PBP2 has a transglycosylase (TGase) domain, too[1]. TPase activity catalyzes cross-linking between peptides carried by two adjacent glycan chains, whereas the TGase activity catalyzes glycan chain elongation from lipid II substrate[1,29]. Therefore, one would predict that ECg treatment decreases the resistance of USA300 LAC not only to β-lactams but also to TGase inhibitor moenomycin[29] because the localization of PBP2 to the division septum appears to be required for its essential TGase activity[30]. Indeed, this was the case, and the minimal inhibitory concentration (MIC) values of moenomycin against USA300 LAC were reduced 8-fold (from 0.25 μg/mL to 0.03125 μg/mL) in the presence of ECg at a concentration of 12 μg/mL (Fig. S5c).

To further examine the effect of PBP2 delocalization on virulence gene expression, we treated WT USA300 LAC cells with ECg and performed Western blot and qRT-PCR experiments. Our results showed that ECg treatment significantly decreases the production of SpA (Fig. S5d) and the transcription levels of *psmα3*, *psmβ1*, and RNAIII (Fig. S5e). We also observed that ECg treatment causes *vraX* overexpression (180-fold increase) (Fig. S5e), which indicates an activation of VraRS TCS (Fig. 3a, b)[21–23]. These results suggest that PBP2 delocalization is responsible, at least in part, for the dysregulation of gene expression caused by TarO inactivation.

## Genetic or chemical disruption of PBP2 mimics the effects of TarO inhibition

To genetically verify the role of PBP2 in virulence gene expression, we generated a USA300-1 strain from the USA300 LAC by placing the *pbp2* in the chromosome under the control of the isopropyl-β-d-thiogalactopyranoside (IPTG)-inducible *spac* promoter (Fig. S5f). USA300 LAC grew normally regardless of IPTG. However, USA300-1 grew normally only in the presence of IPTG, and its growth was significantly inhibited in the absence of IPTG (Fig. S5g). Importantly, we observed that *pbp2* depletion (Fig. S5h) resulted in a 5-fold decrease in the production of SpA (USA300-1 *vs* USA300, in the absence of IPTG) (Fig. S5i), which also causes an overexpression (230-fold increase) of *vraX* (Fig. S5j) and a significant decrease (>5-fold) in the transcription levels of *psmα3* (Fig. S5k). These results indicate that PBP2 depletion mimics the effects of TarO inactivation.

To further examine the role of PBP2 in mediating virulence gene expression, we followed the expression of *vraX* and PSMα genes through a transcriptional fusion with *lacZ* using a disk diffusion assay. This approach provided a concentration gradient, allowing the detection of *vraX-lacZ* and *psmα-lacZ* without a priori knowledge of the time-dependent induction kinetics of the test compounds. As shown, tunicamycin and PBP2-selective β-lactams (i.e., cefotaxime, cefuroxime, and ceftizoxime)[1,14] each strongly induced *vraX-lacZ* expression and reduced *psmα-lacZ* expression over a wide diffusion zone (e.g., concentration gradient) (Fig. 4a). Expectedly, treatment of USA300 LAC with PBP2-selective β-lactams reduced the production of SpA (Fig. 4b, c). Using a disk diffusion assay with reporter strain USA300::*vraX-lux*, we showed that cefuroxime increased the tunicamycin-induction of *vraX-lux* (Fig. 4d). In contrast, tunicamycin failed to increase the cefuroxime-induction of *vraX-lux* (Fig. 4d), indicating that the TPase activity of PBP2 functions downstream of TarO to modulate the regulatory activity of VraRS.

PBP2 has both TPase and TGase activities[1]. To further examine the role of PBP2 in the regulatory activity of VraRS, we treated USA300::*vraX-lux* and USA300::*psmα-lux* cells with TGase inhibitor moenomycin. We observed that moenomycin treatment dramatically increases the expression of *vraX-lux* (Fig. 4e) while it severely decreased the expression of *psmα-lux* (Fig. 4f) and the production of SpA (Fig. 4g–i). Besides PBP2, the *S. aureus* genome encodes four PG TGases, SgtA, SgtB, RodA, and FtsW, that are thought to have a role in cell wall synthesis[32,33]. Previous studies have also shown that SgtB but not SgtA can support the growth of *S. aureus* in the absence of the main TGase PBP2[32]. We also observed that disruption of *sgtB* in *S. aureus* JE2 strain, a plasmid-cured derivative of USA300 LAC, causes a decrease in the production of SpA (Fig. 4j, k) and a 130-fold increase in the mRNA levels of *vraX* (Fig. 4l), which supports the notion that chemical inhibition of PBP2 TGase induces VraRS activation (Fig. 4e–i).

A very low-dose of cefuroxime (0.3 mg/kg, which is equivalent to $1/1707 \times$ MIC) (Table S1) was able to increase the survival of *G. mellonella* larvae infected by USA300 LAC (Fig. S6a). However, cefuroxime failed to do this for those infected by Δ*agrA*Δ*spa* (a double-deletion mutant lacked both *agrA* and *spa*) at such a low dose (Fig. S6a), although the Δ*agrA*Δ*spa* mutant exhibited increased susceptibility to the cefuroxime when compared to the WT USA300 LAC (Fig. S6b). Thus, under the testing concentration, cefuroxime must inhibit the biological function of either *agrA* or *spa*, or both, to block the virulence of *S. aureus* USA300 LAC. Supporting this notion, Δ*agrA*Δ*spa* mutant is attenuated for virulence (Fig. S6a). However, unexpectedly, cefuroxime administration (0.3 mg/kg body weight) was also able to protect *G. mellonella* larvae from Δ*vraR*-mediated killing (Fig. S6a). A simple explanation for the observation is that when compared to WT USA300 LAC, the Δ*vraR* mutant exhibited increased susceptibility to cefuroxime in vivo like in vitro (Fig. 4a and Fig. S6c). Indeed, it is believed that bacterial growth in vivo is the primary requirement for pathogenicity[34].

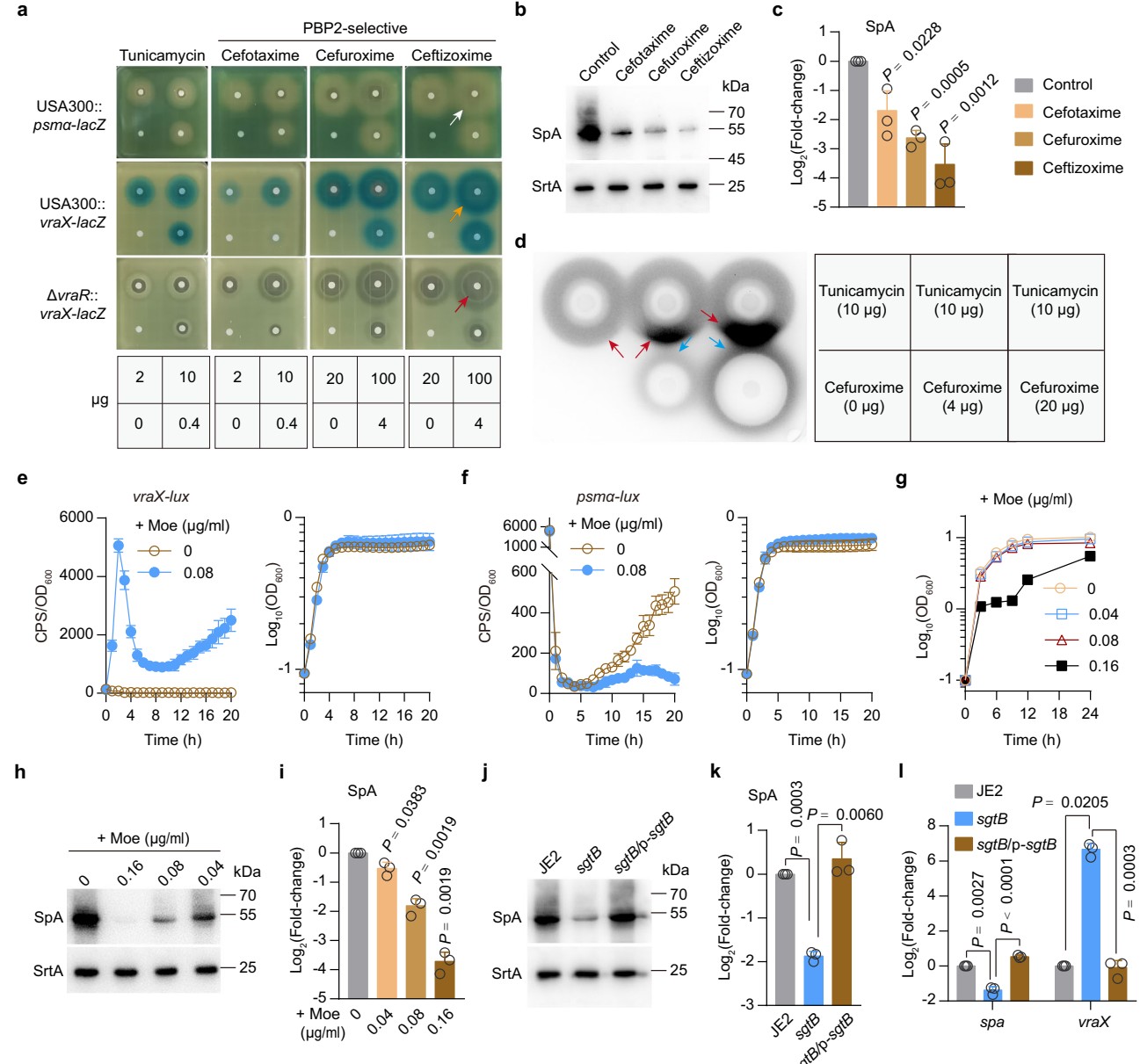

**Fig. 4 | Effects of chemical inhibition of PBP2 or genetic inhibition of SgtB TGase. a** Disk diffusion assay. White arrow indicates a decrease in the expression of *psmα-lacZ*, yellow arrow indicates an increase in the expression of *vraX-lacZ*, and red arrow indicates inhibition zone of bacterial growth, around a paper disk. **b**, **c** Representative images (**b**) and quantitative analysis (**c**) of SpA in *S. aureus* strains grown in TSB medium supplemented with 1.5 μg/ml cefotaxime, cefuroxime, or ceftizoxime for 3 h. Data represent mean ± SD from *n* = 3 independent experiments. **d** Disk diffusion assays for *vraX-lux* induction in USA300 LAC. Red and blue arrows indicate tunicamycin- and cefuroxime-induction of *vraX-lacZ*, respectively. **e**, **f** The expression of *vraX-lux* (**e**) and *psmα-lux* (**f**) in USA300 LAC strain grown in TSB medium supplemented with or without moenomycin (Moe). Data from *n* = 4 biological replicates are reported as the mean ± SD. **g** Effect of moenomycin on the growth of USA300 LAC cultured in TSB medium at 37 °C. Data

represent mean ± SD from *n* = 3 biological replicates. **h**, **i** Representative images (**h**) and quantitative analysis (**i**) of SpA in WT USA300 grown in TSB medium supplemented with or without moenomycin. Data represent mean ± SD from *n* = 3 independent experiments. (**j** and **k**) Representative images (**j**) and quantitative analysis (**k**) of SpA in *S. aureus* strains grown in TSB medium for 3 h. Data represent mean ± SD from *n* = 3 independent experiments. (**l**) qRT-PCR analysis of *spa* and *vraX* in *S. aureus* strains. Data represent mean ± SD from *n* = 3 independent experiments. In **j**–**l**, JE2 and *sgtB* mutant harbor an empty pYJ335 vector as control. Statistical analysis was performed using two-tailed one-sample *t*-test (in **c** and **i**, with the untreated WT USA300 control set to a fold change of 1; in **k** and **l**, when compared with the WT JE2, which was set to a fold change of 1) or otherwise with Student's two-tailed unpaired *t*-test.

We also observed that deletion of *vraR* further attenuated the virulence of Δ*tarO* mutant against *G. mellonella* larvae (Fig. S6d), and the reduced virulence of the Δ*tarO*Δ*vraR* may be related directly to its defects in growth (Fig. S3b). Further studies are needed to understand the mechanisms by which VraR modulate the pathogenicity of *S. aureus*. Nonetheless, these data support the notion that inhibition of PBP2 mimics the effects of TarO inactivation.

## *mecA* plays an important role in mediating the effects of TarO-inhibition

Because PBP2 and PBP2a work in a multienzyme complex and perform a joint action to build the cell wall[1,14,35–37] and because WTA appears to act as a scaffold for the recruitment of PBP2a[5,38,39], we reasoned that PBP2a may be involved in WTA deficiency-induced delocalization of PBP2. To test this hypothesis, we generated a null

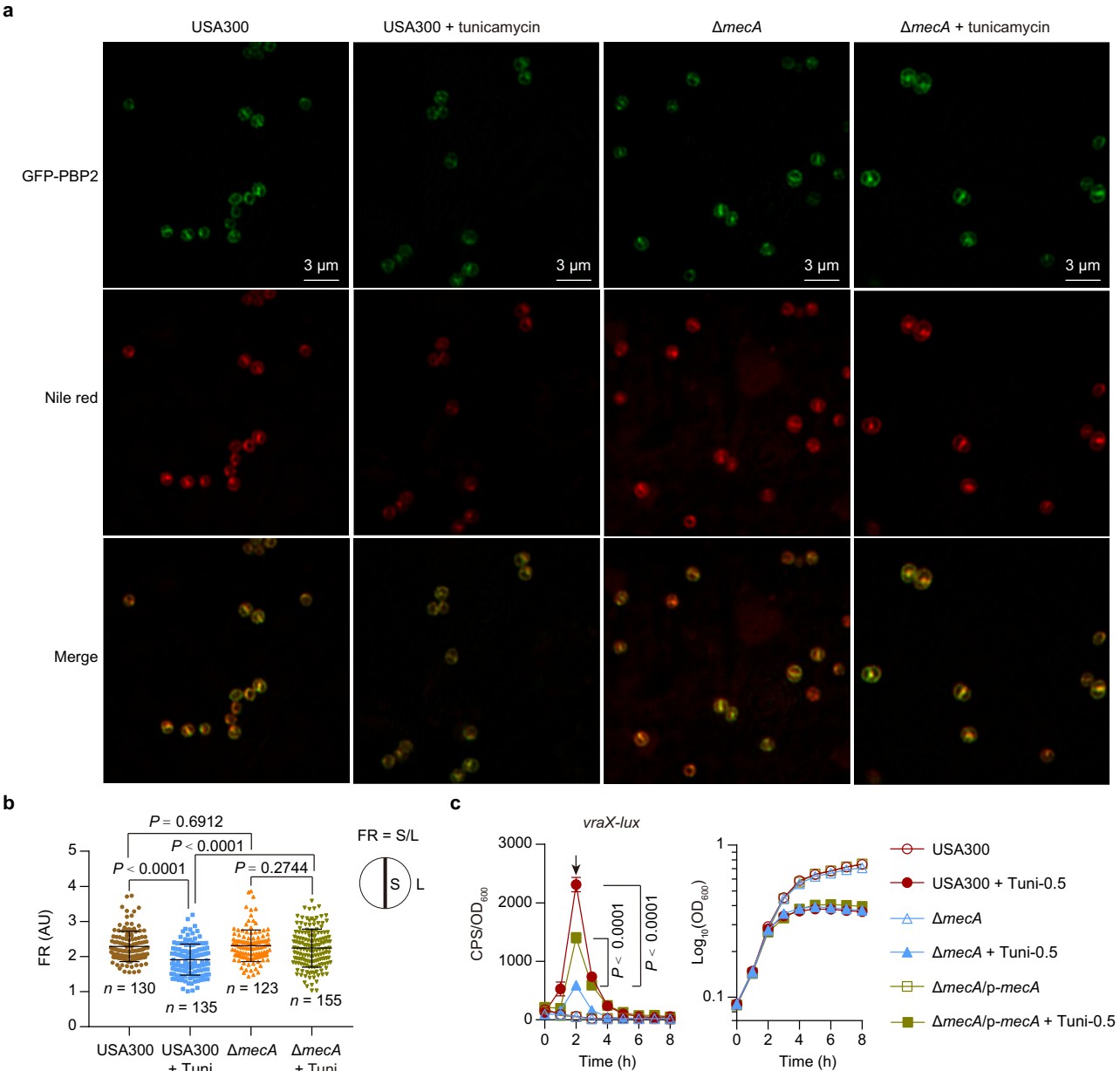

**Fig. 5 | Role of *mecA* in mediating the effects of TarO-inhibition. a** Effect of tunicamycin treatment on the septal localization of GFP-PBP2 in USA300 and its isogeneic mutant Δ*mecA*. *S. aureus* cells expressing GFP-PBP2 fusion proteins were co-stained with lipophilic dye Nile red. The panels (from top to bottom) display fluorescent images of GFP signals, Nile red signals, and dual signals (Merge). The scale bar is 3 μm. **b** Quantitative analysis of GFP-PBP2 fluorescence at septum (S) versus lateral (L) membrane. Data represent mean ± SD. Statistical analysis was performed using unpaired two-tailed Student's *t*-test. **c** Effect of *mecA* deletion on the expression of *vraX-lux* in *S. aureus* strains grown in TSB medium supplemented with either DMSO vehicle control or TarO inhibitor tunicamycin. Tuni-0.5, tunicamycin at a final concentration of 0.5 μg/ml. Data from $n = 4$ biological replicates are reported as the mean ± SD. Statistical analysis was performed at 2 h time point (as indicated by arrow) using Student's two-tailed unpaired *t*-test (compared with tunicamycin-treated Δ*mecA* mutant). USA300 and Δ*mecA* harbors an empty pYJ335 vector as control, respectively.

mutant (Δ*mecA*) that lacks the PBP2a-encoding gene *mecA*. When growth in TSB medium in the absence of tunicamycin, the S/L ratio for GFP-PBP2 in the Δ*mecA* mutant was calculated as 2.31 ± 0.04 ($n = 123$), a value similar to that recorded for the WT USA300 LAC strain (2.29 ± 0.04, $n = 130$) (Fig. 5a, b). As expected, tunicamycin treatment decreased the S/L ratio for GFP-PBP2 (a 16.36% decrease) in the WT USA300 LAC strain (Fig. 5b). A slight decrease in the S/L ratio for GFP-PBP2 was also observed for Δ*mecA* mutant upon tunicamycin treatment (a 2.86% decrease) but this decrease was not statistically significant (Fig. 5b). These results suggest that TarO inhibition-induced delocalization of PBP2 is largely dependent on PBP2a.

To examine whether PBP2a was involved in TarO-inhibition induction of VraRS, we performed promoter fusion analysis again. When grown in the absence of TarO inhibitor tunicamycin, the WT USA300 LAC strain, the Δ*mecA* mutant, and its complemented strain (Δ*mecA*/p-*mecA*) all exhibited very low activity of *vraX-lux* (Fig. 5c), indicating that *mecA* deletion does not induce the regulatory activity of VraRS under our testing conditions. The *vraX-lux* activity in the WT USA300 LAC and the Δ*mecA*-complemented (Δ*mecA*/p-*mecA*) strains are respectively 38- and 27.2-fold increase after 2 h of treatment with 0.5 μg/ml tunicamycin (Fig. 5c). Tunicamycin treatment also can increase the *vraX-lux* activity (11.4-fold increase) in the Δ*mecA* mutant, but to a lesser extent (Fig. 5c). Similar results were observed in

experiments performed with tarocin A1 (Fig. S7a), another specific TarO inhibitor[40]. Thus, *mecA* is important, although not essential, for TarO-inhibition induction of VraRS in USA300 LAC.

To further verify the effect of PBP2a in mediating TarO-inhibition induction of VraRS, we introduced the aTc-inducible *mecA* in a pYJ335 plasmid (p-*mecA*) into RN4220, a MSSA laboratory strain that lacks PBP2a-encoding gene *mecA*. Relative to vehicle control cells (RN4220/pYJ335, RN4220 strain carrying vector control), the RN4220/p-*mecA* strain (RN4220 carrying a plasmid-borne *mecA*) displayed a 2.5-fold higher increase of maximal *vraX-lux* expression when the bacteria were grown in TSB medium supplemented with tunicamycin (Fig. S7b), indicating that leaky expression of *mecA* increases the expression of *vraX-lux*. The *mecA*-induction of *vraX-lux* was even more apparent (a 10-fold induction) in the presence of aTc (Fig. S7c). Similar results were obtained when TarO inhibitor tarocin A1, rather than tunicamycin, was used in the assay (Fig. S7d and e). These data clearly suggest that *mecA* plays an important role in mediating the effects caused by TarO-inhibition. In addition, either tunicamycin or tarocin A1 treatment was able to induce the expression of *vraX-lux* in both USA300 LAC Δ*mecA* mutant (Fig. 5c and Fig. S7a) and WT RN4220 strain (Fig. S7b and d), indicating that some other factor besides *mecA* can assist the TarO-inhibition induction of VraRS in *S. aureus*.

### Blocking Und-P biosynthesis or sequestration of Und-P prevents VraRS activation

Because inhibition of PBP2 caused VraRS activation (Fig. 4)[41], we hypothesize that the accumulation of lipid-linked PG precursors might be a factor for triggering VraRS. To test this possibility, we examined whether clomiphene, an inhibitor of undecaprenyl diphosphate synthase (UPPS) involved in the *de novo* synthesis of carrier lipid Und-P[42] (Fig. 1a), could inhibit the cell-wall-perturbing agents induction of *vraX-lux*. In this assay, a growth-inhibitory concentration of cell-wall-perturbing agents was used in general as the induction kinetics of the *S. aureus vraX* generally correlated inversely with decreasing $OD_{600}$ values[21]. Clomiphene (50 µg/ml) has no inhibitory effect on the expression *vraX-lux* in USA300 LAC cultured in TSB medium (Fig. S8a); however, the clomiphene treatment nearly completely eliminates the maximal moenomycin-induction of *vraX-lux* at 2 h (Fig. 6a). In the presence of clomiphene, moenomycin maximally induced the expression of *vraX-lux* at 8 h, which was ~5.3-fold lower than that seen in the presence of moenomycin alone (2 h) (Fig. 6a). Clomiphene treatment also dramatically reduced the maximal induction of *vraX-lux* by tunicamycin (Fig. 6b) or other cell-wall-perturbing agents (i.e., cefuroxime, vancomycin, fosfomycin, and D-cycloserine) (Fig. S8b–e) that target distinct steps in peptidoglycan biosynthesis (Fig. 1a), ranging from inhibiting the generation of disaccharide–oligopeptide precursors inside the cell (i.e., fosfomycin and D-cycloserine) to preventing the cross-linking of peptide stems outside the cell (i.e., vancomycin and cefotaxime). These results suggest that blocking the *de novo* synthesis of Und-P can prevent the activation of VraRS.

It has been reported that blocking WTA export by targocil causes the accumulation of dead-end lipid-linked WTA intermediates and thus depletes the cellular pool of Und-P available to the biosynthesis of PG[42] (Fig. 1a). We thus examined the effect of targocil treatment on the induction of *vraX-lux*. Like other testing cell wall-acting agents, targocil induced the expression *vraX-lux* at its growth inhibitory concentrations (i.e., 12 µg/ml) under our testing conditions, but the kinetics of induction were slower, and the maximum induction was reached after a culture period of 8 h (Fig. 6c). Of note, targocil treatment (12 µg/ml) nearly completely eliminates the maximal induction of *vraX-lux* by moenomycin (Fig. 6c), vancomycin, fosfomycin, or D-cycloserine (Fig. S9a–c), indicating that sequestration of Und-P prevents VraRS activation. Targocil failed to decrease the tunicamycin-induction of *vraX-lux* (Fig. 6d). In contrast, the addition of tunicamycin (0.5 µg/ml) allowed targocil-treated *S. aureus* cells to express *vraX-lux* similarly to

those treated with tunicamycin alone (Fig. 6d), indicating that accumulation of lipid-linked WTA precursors may have a positive role in the activation of VraRS. Supporting this notion, overexpression of *tarO* was able to significantly increase the expression levels of *vraX-lux* in the USA300 LAC, regardless of the presence of targocil (Fig. S9d and e).

### Overexpression of *murJ* activates VraRS

The final step in the membrane-associated phase of PG synthesis in *S. aureus* involves the translocation of lipid II across the membrane by the flippase MurJ[1,2] (Fig. 1a). We next assessed the impact of *murJ* overexpression on the expression of *vraX-lux*, in order to examine the role of flipped lipid II in the activation of VraRS. When *murJ* was expressed from pYJ335-*murJ*, a multi-copy plasmid with anhydrotetracycline (aTc)-inducible promoter, the *vraX-lux* activity in the WT USA300 LAC strain increased twofold (at 2 h) even in the absence of inducer aTc (Fig. 6e, left panel). The induction of *vraX-lux* became more apparent in presence of 100 ng/ml aTc, and the *murJ* overexpression strain (USA300/pYJ335-*murJ*) displayed a 3.3-fold higher maximal *vraX-lux* expression level compared to its vector control (USA300/pYJ335) at 2 h (Fig. 6e, left panel). Noticeably, *murJ* overexpression had a more profound positive effect on *vraX-lux* expression when TGase inhibitor moenomycin was present (Fig. 6e, right panel). As shown, *murJ* overexpression (induced by aTc) caused an 8.9-fold increase in *vraX-lux* expression in the presence of moenomycin at a concentration of 3 ng/mL (Fig. 6e, right panel). *murJ* overexpression was also able to significantly increase the maximal *vraX-lux* expression in RN4220 (Fig. 6f). These results suggest that accumulation of flipped lipid II molecules may activate VraRS. However, we cannot rule out the possibility that the MurJ protein itself has a role in the activation of VraRS, directly or indirectly.

### Inhibition of TarO causes similar consequences in CA-MRSA strains

As aforementioned, the blockage of WTA biosynthesis reduced the expression of PSM genes and SpA in USA300 LAC (Fig. 1), a dominant CA-MRSA lineage in North America[3]. To further examine the role of WTA in mediating the expression of key virulence genes in CA-MRSA, we treated USA400 MW2, which is a strain belonging to the second most common CA-MRSA lineage in North America[3], and 2011-137, which belongs to the predominant Asian CA-MRSA lineage ST59[43], with TarO inhibitor tunicamycin. Using western blot experiments, we showed that tunicamycin treatment dramatically decreases (> 5-fold) the production of SpA in these strains as it did for USA300 LAC (Fig. S10a and b). Using qRT-PCR experiments, we observed that tunicamycin treatment significantly inhibits the expression of *psmα3*, *psmαβ1*, and RNAIII in these two CA-MRSA strains (Fig. S10c and d). Moreover, increased expression of *vraX* (> 20-fold) was also observed in these two testing *S. aureus* strains upon tunicamycin treatment (Fig. S10c, d). These results indicate that the effect of TarO inhibition on the activity of VraRS TCS and *agr* QS is not limited to the USA300 LAC.

In addition, confirming previous literature reports[13,14], tunicamycin reduced the resistance of both USA400 MW2 and 2011-137 strains to PBP2-selective β-lactams including cefotaxime, cefuroxime, and ceftizoxime (Table S1). Besides, tunicamycin treatment also decreased (4-fold) the MIC values of TGase inhibitor moenomycin against USA300 LAC, USA400 MW2, and 2011-137 strains (Table S1), indicating that disruption of WTA biosynthesis may cause PBP2 delocalization so that a small amount of PBP2-targeting antibiotics is enough to inhibit the residual activity of PBP2 that remain in the correctly localized PBP2 (Figs. 5a and 7).

### Discussion

The dramatic increase in virulent CA-MRSA infections has created a dire need for alternative routes of drug development[3,4,44,45]. In this study, using high-throughput chemical screening, we identified 19

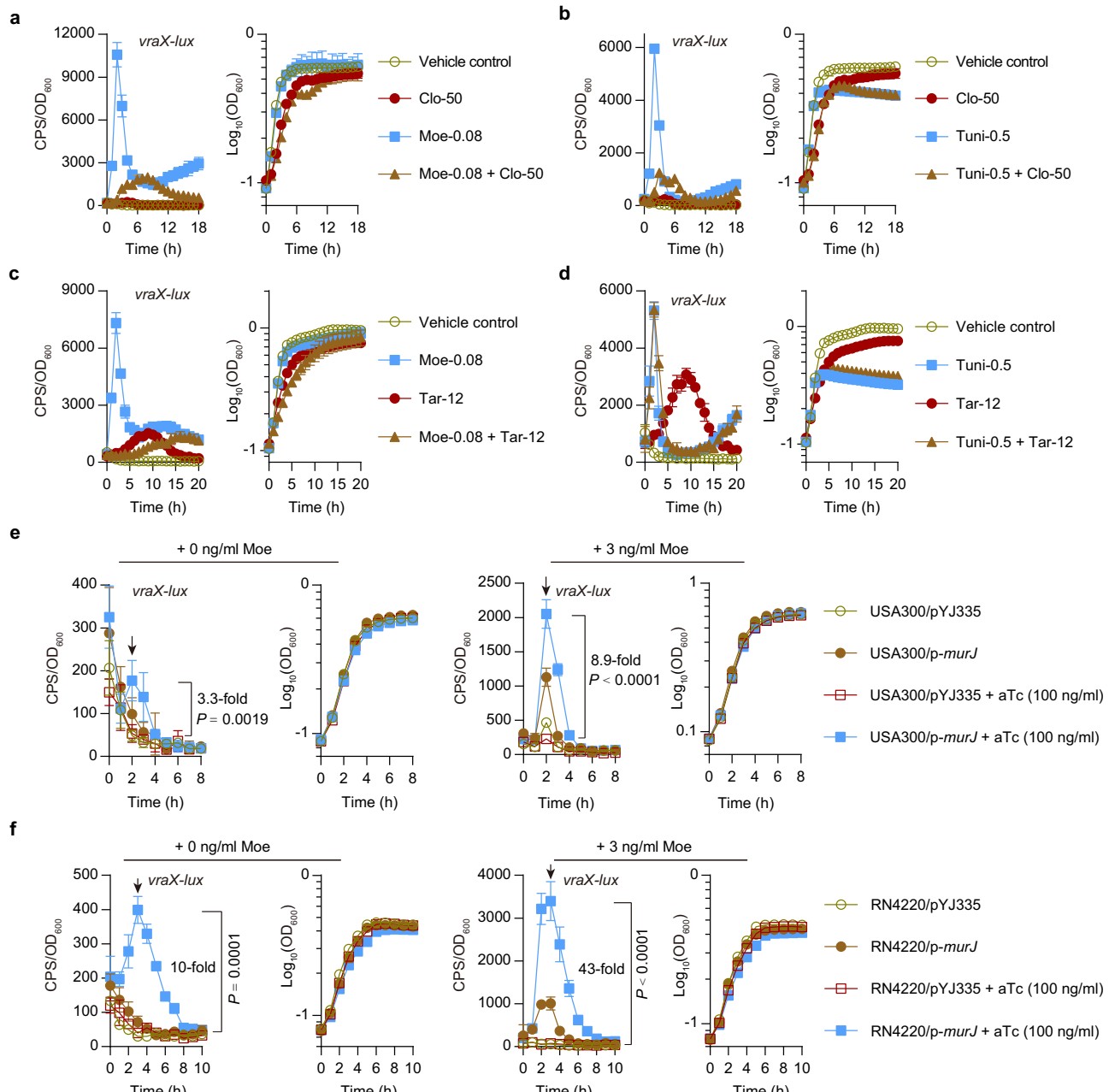

**Fig. 6 | Effect of clomiphene, targocil, and *murJ* overexpression on the expression of *vraX-lux*. a**, **b** Effect of clomiphene treatment on the *vraX-lux* induction by moenomycin (Moe-0.08, at a final concentration of 0.08 µg/ml) (**a**) and tunicamycin (Tuni-0.5; 0.5 µg/ml) (**b**). Clo-50, Clomiphene at a final concentration of 50 µg/ml. Data from *n* = 4 biological replicates are reported as the mean ± SD. **c**, **d** Effect of targocil treatment on the *vraX-lux* induction by moenomycin (Moe-0.08, at a final concentration of 0.08 µg/ml) (**c**) and tunicamycin (Tuni-0.5; 0.5 µg/ml) (**d**). Tar-12, targocil at a final concentration of 12 µg/ml. Data from *n* = 4 biological replicates are reported as the mean ± SD. **e**, **f** Effect of *murJ* overexpression on the expression of *vraX-lux* in USA300 LAC (**e**) and RN4220 (**f**). The expression of *vraX-lux* in USA300 LAC derivatives grown in TSB medium supplemented with or without moenomycin (at a final concentration of 3 ng/ml) in the absence or presence of anhydrotetracycline (aTc) that induces the tetracycline-inducible *xyl/tetO* promoter in pYJ335 plasmid. Data from *n* = 4 biological replicates are reported as the mean ± SD. Statistical analysis was performed using Student's two-tailed unpaired *t*-test (USA300/p-*murJ* versus USA300/pYJ335 at 2 h time point or RN4220/p-*murJ* versus RN4220/pYJ335 at 3 h time point, as indicated by the arrows).

compounds having the potential capacity to reduce the promoter activity of *psmα* operon (Sheet 2 in Supplementary Data 1) that encodes key virulence determinants of *S. aureus*[9,10]. A particularly exciting outcome is the discovery of an unanticipated and important role for WTA biosynthetic pathway in modulating virulence gene expression in CA-MRSA strains (Fig. 7).

WTA is a constituent of cell envelopes with important roles in the physiology and pathogenicity of *S. aureus*[15,16]. We showed here that inhibition of TarO causes VraRS activation and thus a decrease in the

expression of PSM genes and SpA in USA300 LAC (Figs. 1–4, Fig. S3). We determined that the TarO inhibition-induced activation of VraRS was resulted, at least in part, from the loss of functional integrity of the PBP2 *via* a *mecA*-dependent manner (Fig. 5). However, the TarO inhibition-induced activation of VraRS appears to be multifactorial and may involve the redirection of lipid carrier undecaprenyl phosphate (Und-P) flux to the synthesis of lipid-linked peptidoglycan precursors, as both processes compete for the lipid carrier to initiate biosynthesis[1,16] (Figs. 1a and 7).

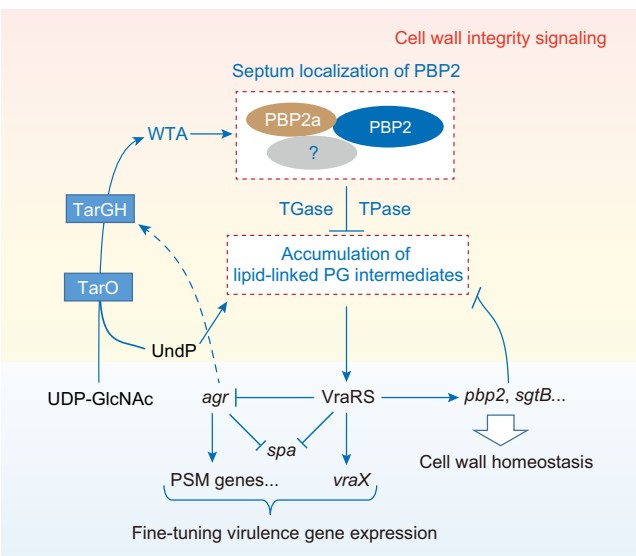

**Fig. 7 | A proposed model for the regulation of virulence gene expression by WTA biosynthesis in CA-MRSA.** The lines show the interaction between the players: arrows indicate activation or generation; solid line indicates a direct influence or direct connection; dotted line indicates the influence in an indirect manner. The *agr* regulatory molecule RNAIII inhibits Rot (Repressor of toxins) translation to derepress the expression of Rot-controlled genes, including *tarH*[8]. WTA biosynthesis may affect the Und-P flux into the lipid II biosynthesis pathway, as both processes compete for the Und-P lipid carrier to initiate biosynthesis.

We showed that inactivation of *tarO* leads to massive dysregulation of key signaling pathways (e.g., VraRS, *agr* QS, and SaeRS) and the resultant dysregulation of numerous virulence genes (Fig. 2, Supplementary Data 2). However, unlike our results, Campbell et al. showed that the effect of tunicamycin on the transcriptome of *S. aureus* COL, an HA-MRSA isolate, was fairly modest[13]. This discrepancy may be due to the utilization of different experimental conditions (i.e., different tunicamycin-treated conditions) or as a consequence of the different genetic backgrounds of the USA300 LAC and COL strains. In MRSA, WTA is believed to act as a scaffold for the recruitment of PBP2a[5,38,39], which also has an important role in localizing *S. aureus* PBP4[19], a key determinant for β-lactam resistance in CA-MRSA (but not in HA-MRSA)[46]. Besides, it was suggested that PBP2, PBP2a, and PBP4 work in a multienzyme complex and perform cooperative actions in building the cell wall[1,14,35–37]. Given these, WTA deficiency may cause mislocalization of either PBP2a or PBP4, or both, in USA300, eventually leading to the concomitant delocalization of PBP2 (Fig. 7). Indeed, *mecA* is important, if not essential, for the TarO inhibition-induction of delocalization of PBP2 (Fig. 5a, b).

Using chemical genetics and the methods of gene overexpression, we provided evidence that the accumulation of flipped lipid II might be a trigger for the activation of VraRS (Fig. 6, Figs. S8 and S9). In bacteria, lipid II is an essential component of the cell walls[47]. The amount of lipid II that can be synthesized is limited, and thus, only a small amount of lipid II is responsible for the fast-growing bacterial cell wall[47]. In our study, transient *vraX-lux* expression is often found when the cell-wall active agents-treated *S. aureus* cells are in an exponential phase of the growth (Fig. 6, Figs. S8 and S9), indicating a role of lipid II accumulation in the activation of VraRS. Paradoxically, like antibiotics that block lipid II utilization (e.g., moenomycin and vancomycin)[48], inhibitors of early steps in peptidoglycan synthesis (e.g., fosfomycin and D-cycloserine) also can induce *vraX-lux* expression (Figs. S8 and S9). A recent study, which was published during the review process of this work, found similar results[49]. One way to explain this paradox is that inhibition of early steps in PG synthesis causes delocalization of PBP2[30],

which in turn will eventually lead to the accumulation of flipped lipid II and thus VraRS activation (Fig. 7). Supporting this notion, it has been reported that antibiotics that blocks lipid II utilization (e.g., oxacillin and vancomycin) trigger an immediate cell wall stress stimulon (CWSS) induction while D-cycloserine and fosfomycin show a lag phase of induction[21]. Moreover, it also has been reported that fosfomycin decreases the expression level of *cwrA*, a VraR-activated gene, after 10 min of treatment while it increases *cwrA* expression after 20–40 min of treatment[50].

Homologs of the VraRS proteins are found in many *Firmicutes* and often mediate an envelope stress response[51]. However, currently, the specific triggers for either VraRS or its homologs are unknown. VraS, the sensor of the VraRS, belongs to a family of intramembrane-sensing histidine kinases (IM-HK) that are characterized by their short input domain, indicative for a sensing process at or from within the membrane interface[51]. It also has been proposed that VraT, which is similar to LiaF encoded in the cell wall-responsive *liaSR* operon that is present in diverse *Firmicutes* bacterial species, is a partner of the VraS[23,51]. Thus, the accumulation of lipid-linked PG intermediates in the cytoplasmic membrane of *S. aureus* might alter the interaction between VraT and VraS, leading to VraRS activation. This hypothesis remains to be determined, however. In addition, we should keep in mind that VraRS activation by cell wall-acting antibiotics may involve a combination of distinct mechanisms or pathways because accumulated lipid-linked WTA precursors appear to activate VraRS (Fig. 6d and Fig. S9d) and because the activity of VraR can be regulated by PknB, which is a lipid II-sensing Ser/Thr kinase[52–54]. Indeed, large differences has been observed in the CWSS induction kinetics of antibiotics when used at MIC levels[21]. Further studies are required to determine the underlying mechanisms of VraRS activation.

CA-MRSA strains often manifest lower levels of oxacillin resistance and are more virulent than HA-MRSA strains[3,6,7]. Studies have also reported that oxacillin acts as an *agr* QS inhibitor to CA-MRSA strains[55,56], and we found a similar result in this study, as evidenced by a decrease in *psma-lacZ* expression levels upon cefotaxime, cefuroxime, or ceftizoxime treatment (Fig. 4a). These previous observations, together with the results of this study, suggest that interaction among WTA biosynthesis, cell wall biosynthesis, VraRS, and *agr* QS may contribute to the success of highly virulent CA-MRSA and that highly virulent CA-MRSA have evolved to optimize virulence gene expression and cell envelope biogenesis upon the acquisition of *mecA* (Fig. 5 and Fig. S7). Our data also suggest that, for the treatment of virulent CA-MRSA infections, WTA biosynthesis inhibitor alone, or in combination with PBP2-selective β-lactams, might have beneficial effects on outcomes by simultaneously attenuating *agr* QS and SpA production (Fig. 7 and Fig. S6a).

In conclusion, in this study we revealed an important role for WTA biosynthesis in the regulation of virulence gene expression in CA-MRSA (Fig. 7). Our results also underscore that chemical genetics is a powerful method for uncovering new biological pathways amenable to pharmacological modulation (Figs. 1a and 7). As maintenance of cell wall homeostasis appears to be important for *S. aureus* infections[57–59], more in-depth knowledge about the role of cell wall integrity signaling pathway in regulating virulence gene expression may help to elucidate the pathogenic success of CA-MRSA and would provide an opportunity to develop novel strategies for the treatment of CA-MRSA infections.

## Methods
### Ethics statement
Mouse infection experiments were performed in strict accordance with the regulations for the Administration of Affairs Concerning Experimental Animals approved by the State Council of People's Republic of China (11-14-1988). The protocols of animal study were reviewed and approved by the Institutional Animal Care and Use Committee (IACUC) of the Shanghai Public Health Clinical Center

(permit 2013P201) and were performed in accordance with the relevant guidelines and regulations. The laboratory animal usage license number is SYXK-HU-2010-0098, certified by Shanghai Committee of Science and Technology.

## Bacterial strains and growth conditions

This study complies with all relevant ethical regulations and the bacterial strains and plasmids used in this study are listed in Table S2. Unless otherwise noted, *S. aureus* strains were grown in tryptic soy broth (TSB, Difco) at 37 °C with shaking (250 rpm), or on tryptic soy agar (TSA, Difco) at 37 °C. *Escherichia coli* strains were grown in Luria-Bertani (LB) broth at 37 °C with shaking (250 rpm) or on LB agar plates at 37 °C. For plasmid maintenance, antibiotics were used at the following concentrations where appropriate: for *S. aureus*, erythromycin (Sangon Biotech) at 10 μg/ml for RN4220 and 80 μg/ml for USA300 LAC and its derivatives, chloramphenicol (Sangon Biotech) at 15 μg/ml; for *E. coli*, carbenicillin (Sangon Biotech) at 150 μg/ml. For other reagents, tunicamycin and vancomycin (Dalian Meilun Biotech Co., Ltd.), targocil (Shanghai TopScience Co, Ltd.), oxacillin (Shanghai Aladdin Biochemical Technology Co.,Ltd.), imipenem, cefuroxime, ceftizoxime, cefaclor, cefoxitin, and epicatechin gallate from MedChemExpress, cefotaxime, isopropyl β-D-1-thiogalactopyranoside (IPTG), and AIP from Sangon Biotech, anhydrotetracycline (aTc) from APExBIO, Tarocin A1 from Merck, moenomycin complex from GlpBio, and nile red from Yeasen Biotechnology Co., Ltd. were used in this study.

## High-throughput screening for *psmα-lacZ* inhibitors

A small molecule library composed of 3987 compounds (Supplementary Data 1) [i.e., 2395 FDA-approved drugs, 1199 clinically tested drugs, and 393 bioactive agents; each in a concentration of 100 μM dissolved in Dimethyl sulfoxide (DMSO)] was screened for *psmα-lacZ* inhibitor on TSA plates using a disk diffusion assay. The screening plates were prepared as follows: 20 ml TSA was added into each 120 mm-120 mm plate until solidification; subsequently, a 25-ml unsolidified TSA (about 45 °C) was mixed with 160 μg/ml 5-Bromo-4-chloro-3-indolyl β-D-galactoside (X-gal) and $5 \times 10^6$ CFU (colony-forming units)/ml (exponential growth phase) *S. aureus* USA300::*psmα-lacZ* and then the mixtures were added into the solidified plate. After cooling, filter paper disks (6 mm in diameter) were placed on top of the agar, and 10 μl of each compound was added onto the paper disk. DMSO was used as a control. The plates were incubated at 37 °C for 24 h for color development. The presence of reduced bluest intensity around the paper disks is an indirect measure of the ability of a compound to inhibit the expression of *psmα-lacZ*. In some cases, the zone of growth inhibition produced by the diffusion of compounds from the filter paper disks was observed. Potential *psmα-lacZ* inhibitors were identified by their ability to obviously reduce the bluest intensity around the filter paper disks (the diameter of the zone of inhibition ≥ 14 mm) but without a visible zone of growth inhibition.

## Construction of *psmα-lacZ* and *vraX-lacZ* transcriptional fusions

The plasmid pCL-*lacZ* carrying a promoterless *lacZ* reporter gene was used to construct promoter-*lacZ* reporter fusion[44,60]. For the construction of *psmα-lacZ*, *psmα* promoter region (−523 to −1 of the start codon) was amplified from *S. aureus* USA300 LAC genomic DNA by PCR using the primers *psmα*-pro-F (with *Eco*RI site) and *psmα*-pro-R (with *Kpn*I site) (Table S3). To generate the *vraX-lacZ*, the promoter region of *vraX* (−565 to −3 of the start codon) was amplified from *S. aureus* USA300 LAC genomic DNA by PCR using the primers *vraX*-pro-F (with *Eco*RI site) and *vraX*-pro-R (with *Kpn*I site) (Table S3). The promoter fragments and the plasmid pCL-*LacZ* were digested with *Eco*RI and *Kpn*I, then ligated into pCL-*LacZ* and introduced to *E. coli* DH5α. The construct was confirmed by DNA sequencing, electroporated into RN4220, then transduced into USA300 LAC or its derivatives using bacteriophage Ø85.

## Construction of gene deletion mutants

For the construction of the *tarO* null mutant (Δ*tarO*), the upstream fragment (-1.1 kb) of the intended deletion was amplified from *S. aureus* USA300 LAC genomic DNA using primers *tarO*-up-F and *tarO*-up-R (Table S3), and the downstream fragment (-1.0 kb) of the intended deletion was amplified with primers *tarO*-down-F and *tarO*-down-R (Table S3). The upstream fragment and the downstream fragment were ligated using primers *tarO*-up-F and *tarO*-down-R via overlapping PCR and the product was used for recombination with plasmid pKOR1[61], yielding pKOR1::*tarO*. The resultant plasmid was introduced to *E. coli* DH5α and sequenced to ensure that no unwanted mutations resulted. Then, the resulting plasmid was transferred by electroporation to *S. aureus* RN4220 and subsequently into *S. aureus* USA300 LAC. The allelic replacement was performed and the deletion of target gene was confirmed by PCR.

A similar strategy was used to construct Δ*vraR*, Δ*mecA*, Δ*agrA*, and Δ*spa* mutants. For the construction of Δ*vraR*, the upstream fragment (-1.0 kb) of the intended deletion was amplified from *S. aureus* USA300 LAC genomic DNA using primers *vraR*-up-F and *vraR*-up-R (Table S3), and the downstream fragment (-1.0 kb) of the intended deletion was amplified with primers *vraR*-down-F and *vraR*-down-R (Table S3). The upstream fragment and the downstream fragment were ligated using primers *vraR*-up-F and *vraR*-down-R via overlapping PCR, and the overlapping PCR product was cloned into plasmid pKOR1, yielding pKOR1::*vraR*. For the construction of Δ*mecA*, the upstream fragment (-1.4 kb) of the intended deletion was amplified from *S. aureus* USA300 LAC genomic DNA using primers *mecA*-up-F and *mecA*-up-R (Table S3), and the downstream fragment (-1.4 kb) of the intended deletion was amplified with primers *mecA*-down-F and *mecA*-down-R (Table S3). The upstream fragment and the downstream fragment were ligated using primers *mecA*-up-F and *mecA*-down-R via overlapping PCR, and the overlapping PCR product was cloned into plasmid pKOR1, yielding pKOR1::*mecA*. For the construction of Δ*agrA*, the upstream fragment (-1.1 kb) of the intended deletion was amplified from *S. aureus* USA300 LAC genomic DNA using primers *agrA*-up-F and *agrA*-up-R (Table S3), and the downstream fragment (-1.2 kb) of the intended deletion was amplified with primers *agrA*-down-F and *agrA*-down-R (Table S3). The upstream fragment and the downstream fragment were ligated using primers *agrA*-up-F and *agrA*-down-R *via* overlapping PCR, and the overlapping PCR product was cloned into plasmid pKOR1, yielding pKOR1::*agrA*. pKOR1::*spa* was also constructed similarly with primer pairs *spa*-up-F/R and down-F/R for the amplification the upstream and the downstream fragment, respectively (Table S3).

For the construction of Δ*tarO*Δ*vraR*, pKOR1::*tarO* was transferred by electroporation to Δ*vraR*. For the construction of Δ*agrA*Δ*spa*, pKOR1::*spa* was transferred by electroporation to Δ*agrA*, respectively, and the allelic replacement was performed as described above.

## Construction of plasmids for gene complementation and gene overexpression

For cloning genes with higher efficiency, we incorporated two restriction enzyme sites (i.e., *Asc*I and *Psp*OMI) into pYJ335 plasmid using site-directed mutagenesis method[62] with primers pYJ335-PCR-F and pYJ335-PCR-R. The resulting plasmid, pYJ335-1, which contains restriction sites for *Eco*RV, *Asc*I, and *Psp*OMI, enables the cloning of DNA fragments with compatible cohesive or blunt ends. The modified plasmid pYJ335-1 was sequenced to ensure that no unwanted mutations resulted. To construct plasmid p-*tarO*, a -1.1 kb DNA fragment containing *tarO* was amplified from *S. aureus* USA300 LAC genomic DNA with primers *tarO*-F and *tarO*-R (Table S3). The *tarO* PCR fragment was digested with *Psp*OMI, and the plamid pYJ335-1 was digested with *Eco*RV and *Psp*OMI, then ligated, where the *tarO* gene was downstream of the tetracycline-inducible *xyl/tetO* promoter of pYJ335-1, yielding plasmid p-*tarO*. Construction of p-*tarO* derivative p-*tarO*$^{G152A}$ was performed with the QuikChange site-directed mutagenesis kit

(StrataGene, catalog no. 200518) and the use of primer pair G152A-F/ G152A-R (Table S3).

To construct plasmid p-*vraR*, a ~0.7 kb DNA fragment containing *vraR* was amplified from *S. aureus* USA300 LAC genomic DNA with primers *vraR*-F and *vraR*-R (Table S3). The *vraR* PCR fragment was digested with *Psp*OMI, and the plamid pYJ335-1 was digested with *Eco*RV and *Psp*OMI, then ligated, where the *vraR* gene was downstream of the tetracycline-inducible *xyl/tetO* promoter of pYJ335-1, yielding plasmid p-*vraR*. To generate plasmid p-*sgtB*, a ~0.85 kb DNA fragment containing *sgtB* was amplified from *S. aureus* JE2 genomic DNA with primers *sgtB*-F and *sgtB*-R (Table S3). The plasmid pYJ335 was digested with *Eco*RV and ligated with the *sgtB* PCR fragment, where *sgtB* gene was in downstream of the tetracycline-inducible *xyl/tetO* promoter of pYJ335, yielding plasmid p-*sgtB*. To generate plasmid p-*mecA*, a ~2.1 kb DNA fragment containing *mecA* was amplified from *S. aureus* USA300 LAC genomic DNA with primers *mecA*-F and *mecA*-R (Table S3). The plasmid pYJ335 was digested with *Eco*RV and ligated with the *mecA* PCR fragment, where *mecA* gene was in downstream of the tetracycline-inducible *xyl/tetO* promoter of pYJ335, yielding plasmid p-*mecA*. To construct plasmid p-*murJ*, a ~1.7 kb DNA fragment containing *murJ* was amplified from *S. aureus* USA300 LAC genomic DNA with primers *murJ*-F and *murJ*-R (Table S3). Then *murJ* PCR fragment was ligated with the plasmid pYJ335 by homologous recombination, where the *murJ* gene was downstream of the tetracycline-inducible *xyl/tetO* promoter of pYJ335, yielding plasmid p-*murJ*. To construct plasmid p-*agrA*, a ~0.76 kb DNA fragment containing *agrA* was amplified from *S. aureus* USA300 LAC genomic DNA with primers *agrA*-F and *agrA*-R (Table S3). The DNA fragment was digested with *Psp*OMI, and the plasmid pYJ335-1 was digested with *Eco*RV and *Psp*OMI, then ligated, where the *agrA* gene was downstream of the tetracycline-inducible *xyl/tetO* promoter of pYJ335-1, yielding plasmid p-*agrA*.

The resulting plasmid was introduced to *E. coli* DH5α and sequenced to ensure that no unwanted mutations resulted. Then the plasmid was transferred by electroporation to *S. aureus* RN4220 and subsequently into *S. aureus* USA300 LAC and its derivatives.

### Construction of *pbp2*-depleted mutant USA300-1

To construct pMutin-HA-*pbp2'* plasmid for the genetic depletion of *pbp2* in USA300 LAC, a DNA fragment covering the ribosome binding site region and the first 630 bp (−20 to +630 of the start codon) of *pbp2* was amplified from USA300 LAC genomic DNA with primers P*spac*-*pbp2*-F and P*spac*-*pbp2*-R (Table S3). The PCR product and the plasmid pMutin-HA[63] were digested with *Hin*dIII and *Kpn*I and then ligated. The recombination plasmid was introduced to *E. coli* DH5α and sequenced to ensure that no unwanted mutations resulted. The resulting plasmid, pMutin-HA-*pbp2'*, was electroporated into RN4220 and clones were selected on TSA plate supplemented with 10 μg/ml erythromycin and 1 mM IPTG. Integration of pMutin-HA-*pbp2'* into the chromosome at the *pbp2* locus of RN4220 was confirmed by PCR. The integrated plasmids were then transduced with Φ85 into USA300 LAC and clones were selected on a TSA plate supplemented with 80 μg/ml erythromycin and 1 mM IPTG, yielding USA300-1 strain.

### Construction of *gfp*-*pbp2* fusion

To construct the *gfp*-*pbp2* fusion, codon-optimized *gfp* gene[64] was synthesized by Generay Biotech and then amplified with primers *gfp*-F and *gfp*-R (Table S3). The *pbp2* gene was amplified from *S. aureus* USA300 LAC genomic DNA with primers *pbp2*-F and *pbp2*-R (Table S3), then the *gfp* fragment and the *pbp2* fragment were ligated using overlapping PCR with primers *gfp*-F and *pbp2*-R. The resulting *gfp*-*pbp2* fragment was digested with *Psp*OMI and the plasmid pYJ335-1 was digested with *Eco*RV and *Psp*OMI, then ligated, where the *gfp*-*pbp2* gene was in downstream of the tetracycline-inducible *xyl/tetO* promoter of pYJ335-1, yielding plasmid p-*gfp*-*pbp2*. The resultant plasmid was introduced to *E. coli* DH5α and sequenced to ensure that no

unwanted mutations resulted. Then, the resulting plasmid was transferred by electroporation to *S. aureus* RN4220, and subsequently into *S. aureus* USA300 LAC, Δ*tarO* and Δ*mecA*, respectively.

### Fluorescence microscopy assay

To detect the localization of GFP-PBP2, overnight cultures of *S. aureus* strains containing the GFP-PBP2 fusion were diluted into 10 ml fresh TSB in a 50-ml tube with initial $OD_{600} \approx 0.1$. Erythromycin (80 μg/ml, final concentration) and anhydrotetracycline (100 ng/ml, final concentration) were also added into the medium to maintain the plasmid and to induce the expression GFP-PBP2 respectively. The diluted cultures were grown for 3 h at 37 °C with shaking (250 rpm) and harvested by centrifugation at $5752 \times g$ for 4 min. The bacteria were washed once and resuspended with 1× sterilized phosphate-buffered saline (PBS) and were further diluted in PBS to final $OD_{600} \approx 0.05$. Each 100 μl bacterial suspension was added onto a glass slide coated with a thin layer of 1% agarose in PBS. Where appropriate, bacteria were stained with nile red (with a final concentration of 16 μg/ml) at room temperature for 10 min with rotation, followed by washing with sterile PBS buffer twice and further diluted in PBS to $OD_{600} \approx 0.05$. Images were obtained by LEICA TCS SP8 confocal microscopy (100× objective, zoom 4.5) and LAS X software. Deconvolution treatment of the images was performed using Huygens Professional software, and then the fluorescence intensity of images was analyzed using Image J software (version 1.4.3.67)[65]. The fluorescence ratio (FR) was determined by dividing the fluorescence intensity at the septum by the fluorescence intensity at the lateral wall and at least 130 cells with closed septa for each group from two independent experiments were chosen to calculate the FR, the resulting data were analyzed by Student's unpaired two-tailed *t*-test.

### WTA extraction and the polyacrylamide gel electrophoresis (PAGE) analysis

Overnight cultures of *S. aureus* strains were diluted into 10 ml fresh TSB, which was supplemented with 80 μg/ml erythromycin or tunicamycin (where appropriate) in a 50-ml tube with initial $OD_{600} \approx 0.1$ and were cultured for 3 h at 37 °C with shaking (250 rpm), then WTAs were extracted[66]. The WTA extractions were subjected to 20% (wt/vol) PAGE in Tris-Glycine running buffer (0.1 M Tris base, 0.1 M Glycine) at 135 V for 120 min and then WTA bands were visualized using the alcian blue-silver staining protocol[67]. Densitometric analysis of gel lanes was executed with the ImageJ software.

### Bacterial growth curve measurement

Overnight cultures of the *S. aureus* strains were diluted into 10 ml fresh TSB supplemented with 80 μg/ml erythromycin or the indicated compound in a 50-ml tube with initial $OD_{600} \approx 0.1$, and the diluted cultures were grown for 24 h at 37 °C with shaking (250 rpm). The growth of bacteria (three biological replicates for each group) was monitored using a nanodrop to measure absorption at 600 nm at the indicated time points. A similar procedure was used to measure the growth curves of USA300-1 and related strains, except that the overnight cultures of the *S. aureus* strains were centrifuged by $5752 \times g$ for 5 min and resuspended into 10 ml fresh TSB to remove the IPTG in the overnight cultures.

### RNA-seq and data analysis

Overnight cultures of WT *S. aureus* USA300 LAC strain and Δ*tarO* mutant were diluted into 10 ml fresh TSB in a 50-ml tube with initial $OD_{600} \approx 0.1$ and were cultured at 37 °C with shaking (250 rpm) for 1.5 h ($OD_{600} \approx 0.6$) or for 3 h ($OD_{600} \approx 3$). The bacteria were harvested by centrifugation, and total RNA was immediately extracted using a Qiagen RNeasy kit (Cat. No. 74104) following the manufacturer's instructions. After rRNA was depleted using the Ribo-off rRNA Depletion kit (bacteria) (Vazyme, Cat#:NR407-2), mRNA was used to generate the

cDNA library according to the VAHTS™ Stranded mRNA-seq Library Prep Kit for Illumina protocol (Vazyme, Cat#:NR601-01) and the quality of cDNA library was analyzed to satisfy the sequencing requirements. Then sequencing was performed using the HiSeq X Ten system (Illumina) completed by Sangon Biotech (Shanghai) Co., Ltd.

Bacterial RNA-seq reads were mapped to the *S. aureus* USA300 LAC genome (GenBank: CP055225.1) using Bowtie 2 (version 2.3.2)[68] with two mismatches allowed. Only uniquely mapped reads were kept for subsequent analyses[69]. Differentially expressed genes (DEGs) were determined using DESeq2 Package[70]. The false discovery rate (FDR) was used to identify the q value threshold in multiple tests. Fold change ≥ 2 and *q*-value ≤ 0.05 were used as a threshold to determine significant DEGs. RNA-Seq experiments were performed with three biological replicates for each strain, and the RNA-Seq data have been submitted to the NCBI Sequence Read Archive (SRA) (https://ncbi.nlm.nih.gov/sra/) under BioProject accession number PRJNA746457 with the following Biosample accessions: SAMN20207198 to SAMN20207221.

### Quantitative reverse transcriptase PCR (qRT-PCR) analysis

For qRT-PCR analysis, overnight cultures of *S. aureus* strains were diluted into 10 ml fresh TSB in a 50-ml tube (OD$_{600}$ ≈ 0.1). The cultures were incubated at 37 °C with shaking (250 rpm) for 1.5 h or 3 h, as indicated. RNA samples were extracted using a Qiagen RNeasy kit following the manufacturer's instructions, and the total RNA sample (2 µg) was treated with gDNA digester (YEASEN, Lot: 11123ES60) and reversely transcribed to synthesize cDNA using the Hifair® II 1st Strand cDNA Synthesis SuperMix for qPCR (gDNA digester plus) (YEASEN, Lot: 11123ES60) with random primers according to the manufacture's recommendation. qRT-PCR was carried out in the Bio-Rad 96 well Real-Time PCR System with 20 µl reaction volume, 2 µL of each cDNA dilution was assayed with the Hieff® qPCR SYBR Green Master Mix (YEASEN, Lot:H7901050) and 300 nM primers following the manufacturer's instructions. Melting curve analysis was performed for verification of product homogeneity. The gene-specific primer pairs used for qRT-PCR for *spa*, *psmα3*, *psmβ1*, RNAIII, *vraX*, *pbp2*, *saeR*, *saeS*, *pmtA* and 16 S rRNA are *spa*-RT-F/*spa*-RT-R, *psmα3*-RT-F/*psmα3*-RT-R, *psmβ1*-RT-F/*psmβ1*-RT-R, RNAIII-RT-F/RNAIII-RT-R, *vraX*-RT-F/*vraX*-RT-R, *pbp2*-RT-F/*pbp2*-RT-R, *saeR*-RT-F/*saeR*-RT-R, *saeS*-RT-F/*saeS*-RT-R, *pmtA*-RT-F/*pmtA*-RT-R and 16S-RT-F/16S-RT-R, respectively (Table S3). The amplicon of 16S rRNA was used as an internal control, and the cDNA sample was diluted when analyzing its expression because of high abundance. Relative expression levels of interest genes were calculated by the relative quantification method ($2^{-\Delta\Delta Ct}$ method)[71] and reported as fold-change.

### Western blot analysis

For western blot analysis of SpA, overnight cultures of *S. aureus* strains were diluted into 10 ml fresh TSB in a 50 ml tube with initial OD$_{600}$ ≈ 0.1. The diluted cultures were incubated for 3 h at 37 °C with shaking (250 rpm), and the bacteria were harvested by centrifugation, washed once with 1× TE buffer (10 mM Tris, 1 mM EDTA, pH = 8.0), and resuspended with 1× TE buffer at a final OD$_{600}$ of 10. Then, 100 µl *S. aureus* suspensions were lysed by lysostaphin (50 µg/ml), and each 100 µl lysed sample was mixed with 25 µl of 5 × SDS loading buffer [250 mM Tris-HCl (pH = 6.8), 10% SDS, 0.5% bromophenol blue, 50% glycerol and 100 mM DTT] and heated at 100 °C for 10 min. Each 10 µl prepared sample was subject to 10% (wt/vol) SDS-PAGE in Tris-Glycine running buffer at 135 V for 90 min, and the proteins on the gel were transferred onto PVDF (Bio-Rad) membranes. The membranes were blocked with 5% (wt/vol) skim milk for 120 min and then incubated with the primary antibody at 4 °C overnight, washed four times for 80 min with TBST buffer [10 mM Tris-HCl (pH = 7.5), 150 mM NaCl, 0.05% Tween 20], followed by incubating with the secondary antibody for 120 min, and washed four times for 80 min with TBST. The resulting

chemiluminescent light was detected by a Tanon-5200 multi (Tanon Science & Technology Co., Ltd.). Primary antibodies anti-SpA (Abcam, 1:5000 dilution) and anti-SrtA (Abcam, 1:2500 dilution) and secondary anti-rabbit IgG conjugated to horseradish peroxidase (HRP) (Cwbio, 1:5000 dilution) were used. Biostep™ Prestained Protein Marker (Tanon) was used as a molecular weight reference. Densitometric analysis of gel lanes was executed with the ImageJ software (version 1.4.3.67).

### Hemolytic activity assay

Overnight cultures of the *S. aureus* strains were diluted into 10 ml fresh TSB in a 50-ml tube with initial OD$_{600}$ ≈ 0.1, and the diluted cultures were cultured for 3 h at 37 °C with shaking (250 rpm) and harvested by centrifugation. Sterilized toothpicks were used to stab the bacteria onto 5% (vol/vol) sheep blood agar plates (Kemajia Microbe Technology Co., Ltd.). Zones of clearance surrounding the bacterial colonies indicated hemolysis and were determined at 30 °C for 24 h and then for 24 h at 4 °C after inoculation.

### Minimum inhibitory concentration (MIC) assays

Minimum inhibitory concentration (MIC) assays were performed as follows. Briefly, the *S. aureus* strains were cultured in 10 ml fresh TSB in a 50 ml tube. The exponential phase bacterial suspension was diluted with Mueller-Hinton II broth (cation-adjusted, BD 212322) to ~5 × 10$^5$ CFU/ml. Each 100 µl bacterial dilution was distributed in 96-well plates, and antibiotics were serially diluted twofold. The 96-well plates were incubated at 37 °C for 24 h. MIC values were defined as the lowest compound concentration to inhibit bacterial growth completely.

### Spot dilution assay

To analyze the resistance of *S. aureus* to moenomycin, a spot dilution test was performed[72] with some modifications. Briefly, overnight cultures of *S. aureus* strains were diluted into 10 ml fresh TSB in a 50 ml tube with initial OD$_{600}$ ≈ 0.1, and grown at 37 °C with shaking (250 rpm) for 3 h. Serial 10-fold dilutions of bacterial suspensions were prepared, and 10 µl of each dilution was spotted onto TSA agar plates containing with or without moenomycin (40 ng/ml). The plates were incubated at 30 °C for 24 h and photographed.

### Construction of promoter-*lux* fusions

To construct the promoter-*lux* fusions, coding sequences of *luxABCDE* operon were amplified from plasmid pAUL-A Tn4001 of *S. aureus* Xen 36 [73] by PCR using the primers *luxABCDE*-F (with *Eco*RI-*Xho*I-*Pml*I site) and *luxABCDE*-R (with *Kpn*I site) (Table S3), *luxABCDE* product was digested with *Eco*RI and *Kpn*I and then ligated with the pCL-*lacZ* plasmid, generating pCL-*lux*.

For the construction of *psmα-lux*, *psmα* promoter region (−523 to −13 of the start codon) was amplified from *S. aureus* USA300 LAC genomic DNA by PCR using the primers *psmα*-pro-F (with *Eco*RI site) and *psmα*-pro-*lux*-R (with *Xho*I site) (Table S3). To generate the *vraX-lux*, the promoter region of *vraX* (−565 to −15 of the start codon) was amplified from *S. aureus* USA300 LAC genomic DNA by PCR using the primers *vraX*-pro-F (with *Eco*RI site) and *vraX*-pro-*lux*-R (with *Xho*I site) (Table S3). The promoter fragments were digested with *Eco*RI and *Xho*I, ligated into pCL-*lux* that was also digested with *Eco*RI and *Xho*I, and then introduced to *E. coli* DH5α. The construct was confirmed by DNA sequencing, electroporated into RN4220, then transduced into USA300 LAC or its derivatives using bacteriophage Ø85.

### Monitoring promoter activity by *lux*-based reporters

Overnight cultures of the *S. aureus* promoter-*lux* reporter strains were diluted to an OD$_{600}$ ≈ 0.2 in fresh TSB supplemented with or without test compounds, as indicated. Then, 100 µl aliquot of the sample was distributed to a 96-well black-wall clear-bottom plate (Costar, Corning Incorporated), and 50 µl saxoline was added to each well to prevent

evaporation. Promoter activities were measured using a Synergy 2 Multi-Mode Microplate Reader (Biotek) at different time points of bacterial growth. A similar procedure was used to measure the effects of tunicamycin on the expression of *psmα-lux* in the presence of exogenous addition of AIP, except that the bioluminescence activity and the $OD_{600}$ were only monitored at 2 h. For the maintenance of plasmid pYJ335 or its derivatives, 120 µg/ml erythromycin was used, where appropriate.

### Expression and purification of His₆-VraR

For the heterogeneous expression of N-terminal 6His-tagged VraR (i.e., His₆-VraR), a ~0.65 kb DNA fragment containing *vraR* was amplified from *S. aureus* USA300 LAC genomic DNA with primers His-*vraR*-F and His-*vraR*-R (Table S3). The *vraR* PCR fragment was digested with *Bam*HI and *Xho*I and then ligated into pET28a, which was also digested with *Bam*HI and *Xho*I, generating pET28a-His₆-*vraR*. The construct was introduced to *E. coli* DH5α and sequenced to ensure that no unwanted mutations resulted. Then, the resulting plasmid was transformed into *E. coli* BL21 (DE3).

For purification of His₆-VraR, 10 ml overnight LB cultures of *E. coli* BL21 (DE3) carrying the plasmid pET28a-His₆-*vraR* were diluted 1:100 into 1 L fresh LB medium supplemented with 50 µg/ml kanamycin and grown at 37 °C with shaking (220 rpm) to an $OD_{600}$ of 0.6. Protein was induced by the addition of 0.5 mM IPTG (Sangon) for 16 h at 16 °C. Cells were harvested by centrifugation and resuspended in 50 ml of buffer A (20 mM Tris-HCl, pH 8.0, 1 mM dithiothreitol (DTT), and 0.5 M NaCl). Then, cells were lysed at 4 °C by sonication and centrifuged at $15,294 \times g$ for 25 min at 4 °C to remove insoluble material and the membrane fraction. The supernatant was loaded onto a HisTrap HP column (GE Healthcare Life Sciences China), equilibrated with buffer A and eluted with a 0–100% gradient of buffer B (50 mM Tris (pH 8.0), 300 mM NaCl, 1 mM DTT, and 250 mM imidazole). Then the collected fractions containing His₆-VraR were loaded onto the HiTrap Desalting 5 × 5 ml (Sephadex G-25 S) (GE Healthcare Life Sciences China) and eluted with buffer A to remove the imidazole. The purified His₆-VraR protein was verified by 10% (w/v) SDS-PAGE followed by Coomassie blue staining, and protein concentration of His₆-VraR was determined using a NanoDrop 2000 (Thermofisher China) by $A_{280nm}$.

### Electrophoretic mobility shift assay (EMSA)

For EMSA experiments, 2 ng/µl DNA fragments and various concentrations of His₆-VraR were mixed with binding buffer [10 mM Tris-HCl (pH 7.5), 50 mM KCl, 1 mM DTT, 5 mM MgCl₂, 0.05% Nonidet® P-40, 2.5% Glycerol, and 50 mM acetyl phosphate]. The reactions in 20 µl volume were incubated at room temperature for 15 min. Following incubation, the mixtures were separated by electrophoresis in a 6% native polyacrylamide gel in 0.5× Tris-borate-EDTA (TBE) buffer at 90 V for 110 min at 4 °C. Then, the gel was stained with GelRed nucleic acid staining solution for 10 min and the DNA bands were visualized by a Tanon-5200 multi (Tanon Science & Technology Co., Ltd.).

DNA probes (i.e., *spa*-p, *vraX*-p, *vraRS*-p, and *coa*-p) were PCR-amplified from *S. aureus* USA300 LAC genomic DNA using the primers listed in Table S3. For *spa*-p, a 414 bp DNA fragment covering the promoter region of *spa* (−414 to −1 of the start codon) was amplified with primer pair EMSA-*spa*-F/EMSA-*spa*-R. For *vraX*-p, a 350 bp DNA fragment covering the promoter region of *vraX* (−350 to −1 of the start codon) was amplified with primer pair EMSA-*vraX*-F/EMSA-*vraX*-R. For *vraRS*-p, a 147 bp DNA fragment covering the promoter region of *vraRS* (−252 to −106 of the start codon of the *vraU* in the *vraUTRS* operon)[23] was amplified with primer pair EMSA-*vraRS*-F/EMSA-*vraRS*-R. For *coa*-p, a 228 bp DNA fragment covering the promoter region of *coa* (−199 to +29 of the start codon) was amplified from *S. aureus* USA300 LAC genomic DNA with primers EMSA-*coa*-F/EMSA-*coa*-R. All PCR products were purified with a purification kit (Omega Bio-tek).

### Dye primer-based DNase I footprinting assay

For the DNase I footprinting assay of *spa* promoter, a 414-bp 6-carboxyfluorescein (FAM)-labeled DNA fragments (−414 to −1 of the start codon) was amplified from *S. aureus* USA300 LAC genomic DNA with primers EMSA-*spa*-F/FAM-*spa*-R. PCR products were purified with a purification kit (Omega Bio-tek). Then, 50 µl reaction mixture containing 300 ng spa promoter DNA, 6 µM His₆-VraR, and binding buffer [10 mM Tris-HCl (pH 7.5), 50 mM KCl, 1 mM DTT, 5 mM MgCl₂, 0.05% Nonidet® P-40, 2.5% Glycerol, and 50 mM acetyl phosphate] was incubated at room temperature for 15 min. Subsequently, 0.05 unit of DNase I was added into the reaction mixture for another 3 min incubation and then stopped the DNase I digestion by adding 90 µl quenching solution (200 mM NaCl, 30 mM EDTA, and 1% SDS). The mixture was extracted with phenol–chloroform-isoamyl alcohol (25:24:1), and the digested DNA fragments were isolated by ethanol precipitation, then dried under vacuum and resuspended in RNase-free water. Next, 5 µl of digested DNA was mixed with 4.9 µl of HiDi formamide and 0.1 µl of GeneScan-500 LIZ size standards (Applied Biosystems). A 3730xl DNA analyser was used to detect the sample, and the results were analysed with GeneMapper software (Applied Biosystems). Thermo Sequenase Dye Primer Manual Cycle Sequencing Kit (ThermoFisher) was used to determine the sequences of the His₆-VraR protection region more precisely after the capillary electrophoresis results of the reactions were aligned, and the corresponding label-free promoter DNA fragment was used as a template for DNA sequencing. Then, electropherograms were analysed with GeneMarker v1.91 (Applied Biosystems).

For the DNase I footprinting assay of *vraX* promoter, a 350-bp FAM -labeled *vraX* promoter DNA (−350 to −1 of the start codon) was amplified from *S. aureus* USA300 LAC genomic DNA with primers EMSA-*vraX*-F/FAM-*vraX*-R, and the following procedures were similar to those of the *spa* promoter DNase I footprinting assay.

### *Galleria mellonella* larvae infection model

Overnight cultures of the *S. aureus* strains were diluted into 10 ml fresh TSB in a 50 ml tube with initial $OD_{600} \approx 0.1$, and the diluted cultures were grown for 3 h at 37 °C with shaking (250 rpm), and harvested by centrifugation at $5752 \times g$, 4 °C for 5 min. The bacterial cells were washed twice and resuspended with pre-cooled (4 °C) PBS. Each about 300 mg *G. mellonella* larvae (Tianjin HuiYuDe Biotechnological Co., Ltd.) was injected with sterile PBS or $5 \times 10^6$ CFU *S. aureus* in a 10-µl volume, then cultivated at 37 °C and observed to analyze the mortality data of *G. mellonella* larvae.

To test the effect of cefuroxime on *G. mellonella* larvae infection model, each *G. mellonella* larvae was injected with the indicated concentrations of cefuroxime in a 10-µl volume, and 1 h later, *G. mellonella* larvae was infected by $5 \times 10^6$ CFU *S. aureus* USA300 LAC in a 10-µl volume, then cultivated at 37 °C and observed to analyze the mortality data. For this experiment, the time of inoculation was designated 0 h.

### Mouse infection models

6–8 weeks old female BALB/c mice (about 18–20 g) were purchased from Shanghai Jiesijie Laboratory Animal Co., Ltd (https://shminxing083596.11467.com/). All mice were housed in the specific pathogen-free facility and were given an ad libitum diet. Dark and light were cycled every 12 h. The ambient temperature was 20–26 °C, and the humidity was 40–60%.

Overnight cultures of WT USA300 LAC strain and Δ*tarO* mutant were diluted into 100 ml fresh TSB in a 250 ml flask with initial $OD_{600} \approx 0.1$, the diluted cultures were grown for 3 h at 37 °C with shaking (250 rpm), and harvested by centrifugation at $5752 \times g$, 4 °C for 5 min. The bacteria were washed twice and resuspended with pre-cooled PBS. 6–8 weeks old female BALB/c mice were intravenously administered with 100 µl bacterial suspension via retroorbital injection. For the abscess formation model, mice were challenged with

either $5 \times 10^6$ or $2 \times 10^7$ CFU of *S. aureus* USA300 LAC or its variant Δ*tarO*. *S. aureus*-infected mice were humanely euthanized 5 days after infection on the basis of IACUC-approved criteria, and kidneys, hearts, and livers were aseptically removed. Organs were homogenized in PBS plus 0.1% Triton X-100 and the homogenates of each organ were serial diluted and spotted onto TSA plates, then cultivated at 30 °C for 24 h and enumerated the CFU. Statistical significance was determined by the Mann–Whitney test (two-tailed). For the lethal challenge model, mice were infected with $1 \times 10^8$ CFU of *S. aureus* USA300 LAC and its variant Δ*tarO* by retroorbital injection, and moribund mice were humanely euthanized according to IACUC guidelines to obtain survival data; mice were observed for ten days, and the log-rank test was used to analyze mortality data.

### Statistical analysis

Statistical analysis was conducted using GraphPad Prism version 7.00 software and the number of independent biological experiments are presented in the figure legends. The log-rank test was used for survival analysis. Two-tailed one-sample *t*-test, Student's two-tailed unpaired *t*-test or Mann–Whitney two-tailed test were used to compare two datasets where appropriate.

### Reporting summary

Further information on research design is available in the Nature Portfolio Reporting Summary linked to this article.

## Data availability

*S. aureus* USA300 LAC genome can be found at National Center for Biotechnology Information with the accession number of CP055225.1. RNA-Seq data have been deposited in the NCBI Sequence Read Archive (SRA), with accession number PRJNA746457. Other data are included in the article and/or the supplementary information and source data file. Source data are provided with this paper.

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

## Acknowledgements

This work was supported by grants from National Natural Science Foundation (NSFC) (grant nos. 32270184 to L.L. and 31870127 to L.L.). This study was also funded by the Science and Technology Commission of Shanghai Municipality (grant no. 19JC1416400 to L.L.) and State Key Laboratory of Drug Research (SIMM2003ZZ-03 to L.L. and SIMM2205KF-07 to L.L.). We appreciate RNA-seq and the corresponding initial data analysis from Sangon Biotech (shanghai) Co., Ltd.

## Author contributions

Y.L. and L.L. conceived and initiated the study. Y.L. and F.C. did most experiments. Q.Z. performed murJ overexpression assays, Q.C. performed DNase I footprinting assays, H.H. performed microscopic analysis. Y.L. F.C., Q.Z, R.C., H.P., Y.W., R.H., Q.L., M.L., T.B., H.L., and L.L. analysed the data. L.L. supervised the study and wrote the manuscript with input from Y.L. and F.C. All authors discussed the results and commented on the manuscript.

## Competing interests

The authors declare no competing interests.
