## [Peer Review File · Nature Communications]

Modulation of MRSA virulence gene expression by the wall teichoic acid enzyme TarOREVIEWER COMMENTS

Reviewer #1 (Remarks to the Author):

In this study, the authors sought to identify inhibitor(s) of psm-a expression, a key virulence factor of CA-MRSA. Surprisingly the high throughput screen identified tunicamycin, an inhibitor of TarO and MraY at low and high concentrations, respectively. The authors provide strong evidence that TarO is the link between tunicamycin and the repression of virulence genes in CA-MRSA. They demonstrate that repression is caused by the activation of the transcriptional regulator, VraR. By combining genetic approaches and the use of known chemical inhibitors, the authors show that the loss of WTA leads to the delocalization of PBP2 which triggers the activation of VraR. The delta-tarO or tunicamycin-mediated delocalization of PBP2 was less pronounced in a *mecA* mutant that does not produce PBP2a suggesting an involvement of PBP2a presumably through its interaction with PBP2.

The text is dense and the authors share many details that are not absolutely necessary for the 'main story' (too many messages). It isn't always easy to follow the rationale although it is clear that the authors try to cover all the bases. The title should state the main finding of the paper more clearly. In summary, this is an interesting study, the findings could be related more succinctly.

Comments:

1. Growth curves should be plotted on a semi-log scale, preferably with log₁₀. Similarly, the fold change of transcripts could be presented on a log scale.
2. Line 186: should this be Fig 2D instead of 2A?
3. Line 187: is the 14.7-fold value derived from Fig 2C or from the Excel file?
4. Line 247: delete 'full' in this title
5. Figure 3E: Does VraR bind 300 bases upstream of the *spa* promoter? If so how does VraR exert its repressor activity?
6. Line 324: the acronym ECg should be introduced and explained
7. Line 367-Supplementary figure 5FG: addition of IPTG to USA300-1 increases PBP2 expression, however it is well below levels observed for USA300 (5G). This is surprising given the fact that IPTG fully restores growth of USA300-1 to wild type levels (5F).
8. Line 383-Fig 4D: this needs to be better explained.
9. Line 403: this entire section needs to be better explained and the role of PBP2a should be stated clearly. The word 'augment' is confusing here. Are any of the data for *vraX*-lux expression statistically different?
10. Overall, the microscopy images could be improved. For example, a cytoplasmic membrane dye could be used to enhance the localization pattern of PBP2.
11. There are many plots overlaid on the figures (e.g., Fig. 5C-F). It is difficult to see some of the plots or even know if they are on the figure. This makes it difficult to interpret the data.

Reviewer #2 (Remarks to the Author):

In 'Chemical genetics unveils WTA/PBP2a-PBP2/VraRS as a regulatory axis for virulence gene expression in CA-MRSA', Lu et al. provide compelling evidence that cell envelope homeostasis is an important signal for virulence factor expression in community acquired strains of methicillin resistant *Staphylococcus aureus*. The authors perform a chemical screen seeking to identify inhibitors of psm virulence expression. The screen revealed that tunicamycin, a wall teichoic acid (WTA) inhibitor, and other beta-lactam antibiotics inhibit psm virulence gene expression. Mechanistic studies using genetic approaches, transcriptomics, and other chemical inhibitors show that cell envelope-targeting antibiotics decrease psm and *spa* expression via induction of VraRS two component system repression. Another major conclusion is the finding that accumulated, flipped lipid II is the signal for VraRS. The study is well-designed, contains the appropriate controls, and the data support the conclusions. Some exceptions are noted below. The findings would be of interest to the Nature Communications audience, a recent publication (Fernandes et al. *J. Bact* 2022, PMID: 35862765) demonstrating that inhibition of PBP2 is the mechanism of VraRS activation slightly limits enthusiasm for this work.

Major criticisms -

1. The model that impaired cell envelope homeostasis alters virulence expression via VraRS could be validated in vivo using the animal models. For example, in Fig. 4E the authors demonstrate that cefuroxime treatment protects *G. mellonella*. Their model predicts this is due to VraR induced repression of virulence. Therefore, it stands to reason that a *vraR* mutant will exhibit increased virulence, and this can be tested by challenging cefuroxime-treated and untreated *G. mellonella* with the *vraR* mutant. Similarly, the *tarO* mutant virulence defects presented in Fig. 2I-K should be suppressed to some degree by inactivation of *vraR* (Fig. 2).
2. Generally speaking, *S. aureus* impaired for WTA production (*tarO* mutant/tunicamycin treated) exhibit a clumping phenotype which impacts the capacity of optical density to accurately quantify proliferation. Quantifying proliferation via serial dilutions and enumeration of CFU will provide better resolution of growth.
3. In line 63, the phrase 'resistant to lower concentrations of β -lactam antibiotics' is difficult to follow.
4. In regards to the *psm alpha 3* and *RNAIII* expression data presented in Fig. 3B and Fig. S3 B and C, the more compelling comparisons are between the tunicamycin treated WT and *vraR* mutant. Are those differences significant? Also, the findings could be validated by quantifying *psm alpha 3* and *RNAIII* transcript levels in the *tarO vraR* double mutant as was performed for *Spa* (Fig. S3E right panel, this was protein levels, not transcript).
5. A control demonstrating that clomiphene does not affect Lux activity should be included in Fig. 6.
6. Alternatively, lux expression could be quantified via qRT-PCR.
6. *S. aureus* is typically thought to be resistant to fosfomycin due to activity of FosB. Its curious that fosfomycin causes decreased growth (as measured by optical density). The authors should provide rationale for using the concentrations of fosfomycin and the other antibiotics used in experiments presented in Fig. 6 and Fig S6.
7. It would be interesting to know how the mechanism of VraSR activation presented by Fernandes et al (J. Bact 2022, PMID: 35862765) is interpreted by the authors given their results.

Minor criticisms -

1. It's difficult to understand the screening schematic presented in Fig. 1A. Based on the descriptions it appears that three potential inhibitors are presented in the figure. Including a positive and negative control would help orientate the audience. Similarly, results presented in figure. 4A are difficult to interpret because it's not clear where bacterial growth begins.
2. Line 324 - ECG should be spelled out and a brief explanation of how it effects cells will increase clarity.
3. Fig. 5F is redundant with Fig. E and can moved to supplemental information.

Reviewer #3 (Remarks to the Author):

This manuscript describes the effects of tunicamycin and several other cell-wall targeting compounds on *S. aureus* virulence. The authors provide good evidence that the aforementioned compounds evoke changes in *S. aureus* virulence through several different pathways, involving the VraRS two component system, the Agr quorum sensing system, WTA biosynthesis and cell wall biosynthesis. Although the effect of tunicamycin on bacterial cells are well known, as well as the effects of all other compounds used in this study, the manuscript adds considerable new knowledge by describing the complex interplay between the involved biosynthesis and regulatory systems.

Comments:

On several occasions (e.g. Figure 1), the authors state that used 3 technical replicates of three independent experiments, and they indicate n=9. Technical replicates and biological replicates cannot be treated as equal. The median of each technical triplicate has to be used as one datapoint, resulting in n = 3 (from the biological replicates).

General: It would be beneficial for the reader if the scheme shown in Fig. 6A would be used in the first figure. This will allow the reader to get a good overview of the major WTA and PG biosynthetic pathways and the different compounds known to interfere with the pathways.

Figure 1F: Can the authors please provide information how the experiment shown in figure 1F was performed. This is not entirely clear based on the material and methods section. Based on the extremely low value for the untreated strain, I assume the authors used a diluted overnight culture, and measure activity after 2 h, which would correspond to the second datapoint shown in Figure 1E, is this correct? Did the authors use the addition of synthetic AIP to correct for the low value of the promoter fusion in the growth phase they chose to use?

Line 175 – 178: Can the authors please explain why they referenced Fig. S2A in the context of this sentence?

Line 215 – 218: could the authors rephrase this sentence? They state that agrD was dramatically downregulated, which is not a quantifiable statement. And looking at certain other genes that are > 20-fold regulated, the word dramatically for an 8-fold downregulated gene seems to be an odd choice.

Lines 218 – 220: The authors speculate that the observed downregulation of PSM genes and the PmtABCD system might be due to downregulation of the agr locus. Looking at the locus, the most strongly regulated of the genes is agrD, which encodes the AIP peptide. The TCS components AgrA and AgrC are not very strongly regulated and therefore could the addition of synthetic AIP rescue PSM and Pmt downregulation?

Line 279 -281: I assume the authors meant psma and not psma operon?

Lines 346 – 349: It would be beneficial for the reader if the authors could include a figure depicting the data that shows increased sensitivity against moenomycin upon ECg treatment.

Line 381 – 385: Can the authors please explain in more detail how they arrive at this conclusion.

Line 386: Can the authors please refer to supplementary Table 1 in this sentence, which shows the MIC value of cefuroxime against USA300 LAC.

Line 413: I think this line should say: "The S/L ratio for GFP-PBP2 was also decreased for"... or else "

A decrease in the S/L ratio for GFP-PBP2 was also observed for"

Lines 430 – 438: Can the authors please clarify if the RN4220 strain has a mecA gene on the chromosome and introduction of p-mecA simply adds a second copy on plasmid. This is not clear to the reader based on the text supplied. RN4220 + Tuni-05 already shows vraX induction, which would be a logical consequence if RN4220 has its own mecA gene or if there are other factors involved in vraX induction besides mecA that have not been identified in this manuscript. In addition, is the used vector leaky? RN4220/p-mecA + Tuni-0.5 shows clear induction in Fig. 5E and F and the induction is clearly higher than RN4220+Tuni-0.5 but also clearly lower than RN4220/p-mecA + aTc-0.1 + Tuni-0.5. all of this would suggest the vector is leaky. If so, can the authors please make statement in the text.

Lines 470 – 473 and Fig. 6E: The authors state that tunicamycin addition rendered targocil inactive and allowed S. aureus cell growth similar to tunicamycin treatment alone. Tunicamycin and tunicamycin + targocil treatment did indeed show the same growth pattern, but they grow worse than non-treated or targocil treated cells. Therefore, the sentence seems to be misleading by suggesting tunicamycin + targocil treatment shows better growth. The sentence should be

reworded

Lines 473-477: this sentence is also confusing for the reader. Based on Fig. S6H, targocil does have an effect on vraX induction, but the effect is not necessarily prolonged, in the case of Tar-3, the maximum induction point is simply time shifted and for Tar-12, there is prolonged induction, but it is overall low and never reaches the strength of induction that is seen with the other two targocil concentrations or when targocil is not present. They should rephrase the sentence.

Line 561: The authors state that they showed that accumulation of flipped lipid II is a trigger for VraRS activation. As far as I can tell, the authors never directly show lipid II accumulation. They presumably make the statement based on the observed activity of the different compounds that block different steps in PG and WTA biosynthesis. They need to be more clear in the text.

Line 576: CWSS (cell wall stress stimulon?) was not introduced as abbreviation

Lines 598 – 600: The authors never directly show accumulation of lipid-linked PG or WTA intermediates. They should rephrase the sentence.

Point-by-point responses to Referees' comments:

Dear reviewers,

First, I would like to express my heartfelt appreciation to all of you for evaluating our work and giving the encouraging feedbacks to improve our work. We have carefully revised the manuscripts according to these invaluable comments. We have made point-by-point responses to each of the reviewers' comments below.

Reviewer #1 (Remarks to the Author):

In this study, the authors sought to identify inhibitor(s) of psm-a expression, a key virulence factor of CA-MRSA. Surprisingly the high throughput screen identified tunicamycin, an inhibitor of TarO and MraY at low and high concentrations, respectively. The authors provide strong evidence that TarO is the link between tunicamycin and the repression of virulence genes in CA-MRSA. They demonstrate that repression is caused by the activation of the transcriptional regulator, VraR. By combining genetic approaches and the use of known chemical inhibitors, the authors show that the loss of WTA leads to the delocalization of PBP2 which triggers the activation of VraR. The delta-tarO or tunicamycin-mediated delocalization of PBP2 was less pronounced in a mecA mutant that does not produce PBP2a suggesting an involvement of PBP2a presumably through its interaction with PBP2.

The text is dense and the authors share many details that are not absolutely necessary for the 'main story' (too many messages). It isn't always easy to follow the rationale although it is clear that the authors try to cover all the bases. The title should state the main finding of the paper more clearly. In summary, this is an interesting study, the findings could be related more succinctly.

Response: We are pleased to know that the reviewer finds our study of interest.

We are also thankful to the reviewer for the very valuable and constructive criticism on our study. By following your comments, we have changed the manuscript's title to "WTA/PBP2a-PBP2/VraRS is a regulatory axis for virulence gene expression in CA-MRSA". We also have carefully revised the manuscript to ensure our argument shines, and we will be happy to edit the text further based on the helpful comments.

Comments:

1. Growth curves should be plotted on a semi-log scale, preferably with log₁₀.

Similarly, the fold change of transcripts could be presented on a log scale.

Response: By following your suggestions, in this revised manuscript, the growth curves were plotted on a semi-log scale with log₁₀ and the fold change of transcripts and protein level was presented in a log₂ scale.

2. Line 186: should this be Fig 2D instead of 2A?

Response: We have corrected it.

3. Line 187: is the 14.7-fold value derived from Fig 2C or from the Excel file?

Response: The 14.7-fold value is derived from the Excel file, and we have corrected it.

4. Line 247: delete 'full' in this title

Response: We have deleted the "full" as your suggestion (line 246 in the revised manuscript).

5. Figure 3E: Does VraR bind 300 bases upstream of the *spa* promoter? If so how does VraR exerts its repressor activity?

Response: We are very sorry about the confusion. In our revised manuscript, "*spa* promoter DNA" has been revised as "upstream of *spa*" (in Fig. 3I).

Response regulator VraR is an activator for cell wall biosynthesis-associated genes including *vraRS*, *pbp2*, *sgtB* and *murZ*. In this study, we showed that VraR binds to the promoter sequence (300 bases upstream of the start codon) and represses the transcription of *spa*. It also has been reported that response regulator RhpR inhibits the transcription of *hrpRS* operon by directly binding to an inverted repeat (IR) element located at approximately 958 bp upstream of the coding region of *hrpR* (1).

Currently, the mechanism by which VraR exerts its repressor activity on the expression of *spa* remains unknown but the issue merits further study.

6. Line 324: the acronym ECg should be introduced and explained

Response: We have changed the "...than ECg-treated cells (S/L = 1.65 ± 0.03)" to "than those treated by epicatechin gallate (ECg) (S/L = 1.65 ± 0.03) (Fig. S5B), an inducer for the delocalization of PBP2 in *S. aureus* (31)." in this revised manuscript (line 339-341).

7. Line 367-Supplementary figure 5FG: addition of IPTG to USA300-1 increases PBP2 expression, however it is well below levels observed for USA300 (5G). This is surprising given the fact that IPTG fully restores growth of USA300-1 to wild type levels (5F).

Response: We highly appreciate the careful reading. The addition of IPTG to USA300-1 partially restored the expression of *pbp2* while it fully restored the growth of USA300-1 to wild type levels. Currently we do not have a good explanation for this discrepancy, except to suggest that the maximal growth of USA300 requires a certain level of expression of the PBP2 genes.

8. Line 383-Fig 4D: this needs to be better explained.

Response: By following your comment and the comments by Reviewer #3, we have re-written this part in the revised manuscript as following (line 396-400): Using a disk diffusion assay with reporter strain USA300::*vraX-lux*, we showed that cefuroxime increased the tunicamycin-induction of *vraX-lux*. In contrast, tunicamycin failed to increase the cefuroxime-induction of *vraX-lux* (Fig. 4D), indicating that the TPase

activity of PBP2 functions downstream of TarO to modulate the regulatory activity of VraRS.

Moreover, in order to present the information clearly, we added arrows into the Fig 4D (also see in the following Fig. R1D in this point-to-point response letter) to indicate the positions of tunicamycin- or cefuroxime-induction of *vraX-lux* and described these in the Figure legend as following (line 1472-1473): Red and blue arrows indicate tunicamycin- and cefuroxime-induction of *vraX-lacZ* around a paper disk, respectively.

Fig. R1. Effects of chemical inhibition of PBP2 or genetic inhibition of SgtB TGase. (D) Disk diffusion assays for *vraX-lux* induction by tunicamycin and cefuroxime in USA300 LAC. Red and blue arrows indicate tunicamycin- and cefuroxime-induction of *vraX-lacZ* around a paper disk, respectively.

9. Line 403: this entire section needs to be better explained and the role of PBP2a should be stated clearly. The word ‘augment’ is confusing here. Are any of the data for *vraX-lux* expression statistically different?

Response: We have rewritten the section. We have removed the word “augment”. Please see details in Line 435-478 in the revised manuscript.

In this revised manuscript, the level of statistical significance (i.e., *p*-value) was shown (as indicated in the Fig. 5C and Fig. S7).

10. Overall, the microscopy images could be improved. For example, a cytoplasmic membrane dye could be used to enhance the localization pattern of PBP2.

Response: We re-performed the experiments as suggested, and we replaced the original results with the new results in Fig. 5 A and B in the revised manuscript (in Page 69) as following (Fig. R2 A and B):

Fig. R2. Role of *mecA* in mediating the effects of TarO-inhibition.

11. There are many plots overlaid on the figures (e.g., Fig. 5C-F). It is difficult to see some of the plots or even know if they are on the figure. This makes it difficult to interpret the data.

Response: We apologize for the inconvenience. In this revised manuscript, we have carefully reconstructed the figures throughout the paper in order to make the presentation clear. For instance, reconstructed the original Fig. 5F as following (Fig. R3) to avoid overlaid plots and to make the presentation clear.

Original Fig. 5F

Fig. S7 D and E in the revised manuscript

Fig. R3. The reconstruction of the original Fig. 5F

Reviewer #2 (Remarks to the Author):

In 'Chemical genetics unveils WTA/PBP2a-PBP2/VraRS as a regulatory axis for virulence gene expression in CA-MRSA', Lu et al. provide compelling evidence that cell envelope homeostasis is an important signal for virulence factor expression in community acquired strains of methicillin resistant *Staphylococcus aureus*. The authors perform a chemical screen seeking to identify inhibitors of psm virulence expression. The screen revealed that tunicamycin, a wall teichoic acid (WTA) inhibitor, and other beta-lactam antibiotics inhibit psm virulence gene expression. Mechanistic studies using genetic approaches, transcriptomics, and other chemical inhibitors show that cell envelope-targeting antibiotics decrease psm and *spa* expression via induction of VraRS two component system repression. Another major conclusion is the finding that

accumulated, flipped lipid II is the signal for VraRS. The study is well-designed, contains the appropriate controls, and the data support the conclusions.

Some exceptions are noted below. The findings would be of interest to the Nature Communications audience, a recent publication (Fernandes et al J. Bact 2022, PMID: 35862765) demonstrating that inhibition of PBP2 is the mechanism of VraRS activation slightly limits enthusiasm for this work.

Response: We thank the Reviewer for his/her supportive comments and we hope the responses below address the reviewers outstanding concerns.

Major criticisms -

1. The model that impaired cell envelope homeostasis alters virulence expression via VraRS could be validated *in vivo* using the animal models. For example, in Fig. 4E the authors demonstrate that cefuroxime treatment protects *G. mellonella*. Their model predicts this is due to VraR induced repression of virulence. Therefore, it stands to reason that a *vraR* mutant will exhibit increased virulence, and this can be tested by challenging cefuroxime-treated and untreated *G. mellonella* with the *vraR* mutant. Similarly, the *tarO* mutant virulence defects presented in Fig. 2I-K should be suppressed to some degree by inactivation of *vraR* (Fig. 2).

Response: Thank you for the deep insight and the excellent suggestions.

By following your comments, we performed additional *G. mellonella* infection experiments and bacterial growth assays, and the results are showed in Fig. S6 in the revised manuscript.

Based on these new results, we have added a paragraph to the revised manuscript as following (line 414-433): “A very low-dose of cefuroxime (0.3 mg/kg, which is equivalent to $1/1707 \times \text{MIC}$) (Table S1) was able to increase the survival of *G. mellonella* larvae infected by USA300 LAC (Fig. S6A). However, cefuroxime failed to do this for those infected by $\Delta\text{agrA}\Delta\text{spa}$ (a double-deletion mutant lacked both *agrA* and *spa*) at such a low dose (Fig. S6A), although the $\Delta\text{agrA}\Delta\text{spa}$ mutant exhibited increased susceptibility to the cefuroxime when compared to the WT USA300 LAC (Fig. S6B). Thus, under the testing concentration, cefuroxime must inhibit the biological function of either *agrA* or *spa*, or both, to block the virulence of *S. aureus* USA300 LAC. Supporting this notion, $\Delta\text{agrA}\Delta\text{spa}$ mutant is attenuated for virulence (Fig. S6A). However, unexpectedly, cefuroxime administration (0.3 mg/kg body weight) was also able to protect *G. mellonella* larvae from ΔvraR -mediated killing (Fig. S6A). A simple explanation for the observation is that when compared to WT USA300 LAC the ΔvraR mutant exhibited increased susceptibility to cefuroxime *in vivo* like *in vitro* (Fig. 3A, Fig. 4A and Fig. S6C). Indeed, it is believed that bacterial growth *in vivo* is the primary requirement for pathogenicity (2). We also observed that deletion of *vraR* further attenuated the virulence of ΔtarO mutant against *G. mellonella* larvae (Fig. S6D), and the reduced virulence of the $\Delta\text{tarO}\Delta\text{vraR}$ may be related directly to its defects in growth (Fig. S3B). Further studies are needed to understand the mechanisms by which VraR modulate the pathogenicity of *S. aureus*. Nonetheless, these data support the notion that inhibition of PBP2 mimics the effects of TarO inactivation.”.

2. Generally speaking, *S. aureus* impaired for WTA production (*tarO* mutant/tunicamycin treated) exhibit a clumping phenotype which impacts the capacity of optical density to accurately quantify proliferation. Quantifying proliferation via serial dilutions and enumeration of CFU will provide better resolution of growth.

Response: By following your comments, we quantified the proliferation of *S. aureus*, including the wild-type USA300 LAC, $\Delta tarO$ mutant, and its complementation strain (i.e., $\Delta tarO/p-tarO$) via OD₆₀₀ measurement and serial dilution methods. These new results were included in the Fig. S2 A and B in our revised manuscript.

The correlation between OD₆₀₀ measurement and CFU/mL quantification for the growth of $\Delta tarO$ is strong ($R^2 = 0.8978$) while it was lower than that observed for either wild-type USA300 LAC ($R^2 = 0.9359$) or $\Delta tarO/p-tarO$ ($R^2 = 0.9882$).

3. In line 63, the phrase ‘resistant to lower concentrations of β -lactam antibiotics’ is difficult to follow.

Response: We corrected the sentence as “Unlike most strains of HA-MRSA, CA-MRSA strains generally display low-level resistance to β -lactam antibiotics and exhibit higher virulence and often cause disease in otherwise healthy individuals” (line 63-65) in the revised manuscript.

4. In regards to the *psm alpha 3* and *RNAIII* expression data presented in Fig. 3B and Fig. S3 B and C, the more compelling comparisons are between the tunicamycin treated WT and *vraR* mutant. Are those differences significant?

Response: We compared the expression data of *psm α 3* and *RNAIII* as suggested, and the results are shown in Fig. R4 (as following). The data suggest that tunicamycin-inhibition of *psm α 3*, at least in part, depends on *vraR*. In the case of *RNAIII*, we observed similar result but it is not statistically significant. The *psm α 3* data are included in Fig. S3A in this revised manuscript.

Fig. R4. qRT-PCR analysis of the effect of tunicamycin (0.5 μ g/ml) treatment on the expression of *psm α 3* and *RNAIII* in *S. aureus* strains grown in TSB medium for 3 h. Results are showed as relative expressions (log₂ fold changes) compared with the untreated control. Data represent mean \pm SD from three independent experiments. Statistical analysis was performed using Student's two-tailed unpaired t-test (*P < 0.05, **P < 0.01).

Also, the findings could be validated by quantifying psm alpha 3 and RNAIII transcript levels in the *tarO vraR* double mutant as was performed for Spa (Fig. S3E right panel, this was protein levels, not transcript).

Response: By following your comments, we performed qRT-PCR experiments as suggested and these new results were included in Fig. S3 E to G in our revised manuscript, and we added a paragraph to the revised manuscript (line 293-306) as following: “To further examine the role of VraR in mediating the effects of deletion of *tarO*, we performed qRT-PCR analyses of *spa*, *psmA*, and RNAIII for WT USA300 LAC strain and its derivative strains (e.g., $\Delta tarO$, $\Delta tarO\Delta vraR$, and $\Delta tarO\Delta vraR$ complemented with plasmid-borne *vraR*) grown in TSB medium for 3 h. In this assay, RNA from $\Delta tarO\Delta vraR$ mutant cells of prolonged culture (i.e., at 4 h and 5 h after incubation) were also included because this mutant exhibited defects in growth and because *agr* system regulates *spa*, *psmA*, and RNAIII by responding to cell population density (3, 4). When grown in TSB medium for 5 h, the $\Delta tarO\Delta vraR$ mutant reached a population density ($OD_{600} \approx 2.4$) comparable to the population density of other tested strains cultured for 3 h and exhibited higher expression level of *spa*, *psmA*, and RNAIII than the $\Delta tarO$ mutant (Fig. S3 E to G). Ectopic expression of the *vraR* gene in $\Delta tarO\Delta vraR$ mutant resulted in reduced levels of *spa*, *psmA*, and RNAIII as observed for $\Delta tarO$ mutant (Fig. S3 E to G). Thus, VraR inhibits the expression of *spa*, *psmA*, and RNAIII in the $\Delta tarO$ mutant.”.

5. A control demonstrating that clomiphene does not affect Lux activity should be included in Fig. 6. Alternatively, lux expression could be quantified via qRT-PCR.

Response: We included the clomiphene control data in Fig. S8A in the revised manuscript, and we added a statement to the revised manuscript (line 488-490) as following: “Clomiphene (50 $\mu\text{g/ml}$) has no inhibitory effect on the expression *vraX-lux* in USA300 LAC cultured in TSB medium (Fig. S8A); however,....”.

6. *S. aureus* is typically thought to be resistant to fosfomycin due to activity of FosB. Its curious that fosfomycin causes decreased growth (as measured by optical density).

Response: In this study, we showed that at a concentration of 25 $\mu\text{g/ml}$, fosfomycin decreases the growth of USA300 LAC in TSB medium. Our result is consistent with previous studies showing that the MIC values of the fosfomycin tested for *S. aureus* including MRSA strains were $\leq 64 \mu\text{g/mL}$ (5, 6).

The authors should provide rationale for using the concentrations of fosfomycin and the other antibiotics used in experiments presented in Fig. 6 and Fig S6.

Response: By following your comments, we have added a sentence to the revised manuscript (line 486-488) as following: “In this assay, a growth-inhibitory concentration of cell-wall-perturbing agents was used in general as the induction kinetics of the *S. aureus vraX* generally correlated inversely with decreasing OD_{600} values (7).”.

7. It would be interesting to know how the mechanism of VraSR activation presented by Fernandes et al (J. Bact 2022, PMID: 35862765) is interpreted by the authors given their results.

Response: Thank you for pointing these out. In the revised manuscript, we have cited this recent publication (J Bacteriol. 2022, PMID: 35862765) as following (line 613-614): “A recent study, which was published during the review process of this work, found similar results (49)”.

In the study (J Bacteriol. 2022, PMID: 35862765), Fernandes *et al.* showed that all tested cell wall-targeting antibiotics, including those blocking the early (e.g., fosfomycin) or the late (e.g., vancomycin) stages of PG synthesis pathway, can trigger the VraRS system. Based on these observations, Fernandes *et al.* concluded that the signal sensed by VraRS is not an intermediate in the peptidoglycan synthesis pathway.

We obtained similar experimental results in this study. However, we also further found that: 1) either blocking Und-P biosynthesis or sequestration of Und-P prevents the activation of VraRS by cell wall-targeting antibiotics, 2) overexpression of *murJ* activates VraRS. These data support that the accumulation of lipid-linked peptidoglycan precursors may be a trigger for the activation of VraRS, and thus we discussed this in more details in the Discussion section (Line 602-640).

Additionally, Fernandes *et al.* suggest that a decrease in transglycosylase activity leads to shorter glycans and therefore to an increase in the number of glycan strand extremities, which could be sensed by VraRS. However, we do not think this is the most likely explanation for the mechanism of VraSR activation, because inactivation of *sagB*, which results in longer glycan strands in peptidoglycan (8), led to the activation of VraRS system as well (Fig. R4).

Currently, it still remains an open question how VraRS to be activated in *S. aureus*, and studies are undergoing in our laboratory.

Again, we thank you so much for the constructive suggestions in our studies toward VraRS.

Fig. R5. Effect of *sagB* inactivation on the expression of *vraX-lux* in JE2 strain grown in TSB medium. Data from $n = 3$ biological replicates are reported as the mean \pm SD. *sagB*::Tn denotes a JE2 mutant with mariner-based transposon insertion at 357 bp after the start codon of *sagB*. pYJ335-Tc denotes a pYJ335 derivate (i.e., pYJ335 carrying a tetracycline resistance cassette, *tetK*).

Minor criticisms –

1. It's difficult to understand the screening schematic presented in Fig. 1A. Based on the descriptions it appears that three potential inhibitors are presented in the figure. Including a positive and negative control would help orientate the audience.

Response: By following your comments and the comments from Reviewer #3, we have removed the original “Fig. 1A” from the revised manuscript.

Similarly, results presented in figure. 4A are difficult to interpret because it's not clear where bacterial growth begins.

Response: We have reconstructed the Fig. 4A as following (Fig. R6A) in order to present the information clearly.

Fig. R6. Effects of chemical inhibition of PBP2 or genetic inhibition of SgtB TGase (A) Disk diffusion assay for expression of *vraX-lacZ* and *psma-lacZ*. Disks containing indicated antibiotics (at indicated amounts) were placed on the surfaces of TSA plates (containing X-gal) lawn with reporter strain. Plates were incubated at 37°C for 24 h for colour development. White arrow indicates a decrease in the expression of *psma-lacZ*, yellow arrow indicates an increase in the expression of *vraX-lacZ*, and red arrow indicates inhibition zone of bacterial growth, around a paper disk.

2. Line 324 – ECg should be spelled out and a brief explanation of how it effects cells will increase clarity.

Response: In this revised manuscript, we changed the “...than ECg-treated cells (S/L = 1.65 ± 0.03)” to “than those treated by epicatechin gallate (ECg) (S/L = 1.65 ± 0.03) (Fig. S5B), an inducer for the delocalization of PBP2 in *S. aureus* (9).”(line 339-341).

3. Fig. 5F is redundant with Fig. E and can moved to supplemental information.

Response: We have moved the “Fig. 5F” to supplemental information in the revised manuscript.

Reviewer #3 (Remarks to the Author):

This manuscript describes the effects of tunicamycin and several other cell-wall targeting compounds on *S. aureus* virulence. The authors provide good evidence that the aforementioned compounds evoke changes in *S. aureus* virulence through several different pathways, involving the VraRS two component system, the Agr quorum sensing system, WTA biosynthesis and cell wall biosynthesis. Although the effect of tunicamycin on bacterial cells are well known, as well as the effects of all other compounds used in this study, the manuscript adds considerable new knowledge by describing the complex interplay between the involved biosynthesis and regulatory systems.

Response: We are very grateful to the reviewer for his/her positive assessment for our manuscript. Below we provide responses for each point.

Comments:

On several occasions (e.g. Figure 1), the authors state that used 3 technical replicates of three independent experiments, and they indicate n=9. Technical replicates and biological replicates cannot be treated as equal. The median of each technical triplicate has to be used as one data point, resulting in n = 3 (from the biological replicates).

Response: Thank you for pointing this out. We have corrected these issues in this revised manuscript.

General: It would be beneficial for the reader if the scheme shown in Fig. 6A would be used in the first figure. This will allow the reader to get a good overview of the major WTA and PG biosynthetic pathways and the different compounds known to interfere with the pathways.

Response: We agree. We have adjusted the sequence of Figures as suggested.

Figure 1F: Can the authors please provide information how the experiment shown in figure 1F was performed. This is not entirely clear based on the material and methods section. Based on the extremely low value for the untreated strain, I assume the

authors used a diluted overnight culture, and measure activity after 2 h, which would correspond to the second data point shown in Figure 1E, is this correct? Did the authors use the addition of synthetic AIP to correct for the low value of the promoter fusion in the growth phase they chose to use?

Response: We sincerely thank the reviewer for careful reading and we are sorry for the inconvenience.

You are right. In the “Figure 1F” we measured the activity of after 2 h, which corresponds to the second data point shown in “Figure 1E”.

By following comment, we added a sentence to the material and methods section in this revised manuscript as following (line 998-1000): “A similar procedure was used to measure the effects of tunicamycin on the expression of *psmA-lux* in the presence of exogenous addition of AIP, except that the bioluminescence activity and the OD600 were only monitored at 2 h.”

Line 175 – 178: Can the authors please explain why they referenced Fig. S2A in the context of this sentence?

Response: We have corrected the inappropriate figure referenced.

Line 215 – 218: could the authors rephrase this sentence? They state that *agrD* was dramatically downregulated, which is not a quantifiable statement. And looking at certain other genes that are > 20-fold regulated, the word dramatically for an 8-fold downregulated gene seems to be an odd choice.

Response: We agree. We have revised the “dramatically” as “significantly”.

Lines 218 – 220: The authors speculate that the observed downregulation of PSM genes and the PmtABCD system might be due to downregulation of the *agr* locus. Looking at the locus, the most strongly regulated of the genes is *agrD*, which encodes the AIP peptide. The TCS components *AgrA* and *AgrC* are not very strongly regulated and therefore could the addition of synthetic AIP rescue PSM and Pmt downregulation?

Response: We highly appreciate the careful reading. We have performed additional experiments as suggested. These new results were included in the Fig. S2H in our revised manuscript, showing that the addition of synthetic AIP can rescue PSM and Pmt downregulation. We have added a sentence to the revised manuscript as following (line 237-241): “Additionally, we observed that synthetic AIP is able to rescue the *tarO* deletion-induced downregulation of *psmB1* and *pmtA* (Fig. S2H). A similar result was also obtained when the expression levels of *psmA3* were examined, although the increase was not statistically significant (Fig. S2H).”.

Line 279 -281: I assume the authors meant *psmA* and not *psmA* operon?

Response: We sincerely thank the reviewer for careful reading, and we have corrected the “*psmA* operon” as “*psmA*” in this revised manuscript.

Lines 346 – 349: It would be beneficial for the reader if the authors could include a

figure depicting the data that shows increased sensitivity against moenomycin upon ECg treatment.

Response: Thank you for your advice. By following your suggestion, we have included a figure (Fig. S5C) to depict the data in this revised manuscript.

Line 381 – 385: Can the authors please explain in more detail how they arrive at this conclusion.

Response: By following your comment and the comments by **Reviewer #1**, we have re-written this part in the revised manuscript as following (line 396-400): “Using a disk diffusion assay with reporter strain USA300::*vraX-lux*, we showed that cefuroxime increased the tunicamycin-induction of *vraX-lux*. In contrast, tunicamycin failed to increase the cefuroxime-induction of *vraX-lux* (Fig. 4D), indicating that the TPase activity of PBP2 functions downstream of TarO to modulates the regulatory activity of VraRS.”

Moreover, in order to present the information clearly, we added arrows into the Fig 4D to indicate the positions of tunicamycin- or cefuroxime-induction of *vraX-lux* and we described these in the Figure legend as following (line 1469-1470): Red and blue arrows indicate tunicamycin- and cefuroxime-induction of *vraX-lacZ* around a paper disk, respectively.

Line 386: Can the authors please refer to supplementary Table 1 in this sentence, which shows the MIC value of cefuroxime against USA300 LAC.

Response: We have referred to supplementary Table 1 as suggested.

Line 413: I think this line should say: “The S/L ratio for GFP-PBP2 was also decreased for” ... or else “A decrease in the S/L ratio for GFP-PBP2 was also observed for”

Response: Thank you for pointing this out. We have corrected the expression error.

Lines 430 – 438: Can the authors please clarify if the RN4220 strain has a *mecA* gene on the chromosome and introduction of p-*mecA* simply adds a second copy on plasmid. This is not clear to the reader based on the text supplied. RN4220 + Tuni-05 already shows *vraX* induction, which would be a logical consequence if RN4220 has its own *mecA* gene or if there are other factors involved in *vraX* induction besides *mecA* that have not been identified in this manuscript. In addition, is the used vector leaky? RN4220/p-*mecA* + Tuni-0.5 shows clear induction in Fig. 5E and F and the induction is clearly higher than RN4220+Tuni-0.5 but also clearly lower than RN4220/p-*mecA* + aTc-0.1 + Tuni-0.5. all of this would suggest the vector is leaky. If so, can the authors please make statement in the text.

Response: We highly appreciate the careful reading and thank you for pointing this out. By following your comments, we revised the paragraph as following (Line 463-478): “To further verify the effect of PBP2a in mediating TarO-inhibition induction of VraRS, we introduced the aTc-inducible *mecA* in a pYJ335 plasmid (p-*mecA*) into RN4220, a MSSA laboratory strain that lacks PBP2a-encoding gene *mecA*. Relative to vehicle control cells (RN4220/pYJ335, RN4220 strain carrying vector control), the

RN4220/p-*mecA* strain (RN4220 carrying a plasmid-borne *mecA*) displayed a 2.5-fold higher increase of maximal *vraX-lux* expression when the bacteria were grown in TSB medium supplemented with tunicamycin (Fig. S7B), indicating that leaky expression of *mecA* increases the expression of *vraX-lux*. The *mecA*-induction of *vraX-lux* was even more apparent (a 10-fold induction) in the presence of aTc (Fig. S7C). Similar results were obtained when TarO inhibitor tarocin A1, rather than tunicamycin, was used in the assay (Fig. S7 D and E). These data clearly suggest that *mecA* plays an important role in mediating the effects caused by TarO-inhibition. Additionally, either tunicamycin or tarocin A1 treatment was able to induce the expression of *vraX-lux* in both USA300 LAC Δ *mecA* mutant (Fig. 5C and Fig. S7A) and WT RN4220 strain (Fig. S7 B and D), indicating that some other factor besides *mecA* can assist the TarO-inhibition induction of VraRS in *S. aureus*.”.

Lines 470 – 473 and Fig. 6E: The authors state that tunicamycin addition rendered targocil inactive and allowed *S. aureus* cell growth similar to tunicamycin treatment alone. Tunicamycin and tunicamycin + targocil treatment did indeed show the same growth pattern, but they grow worse than non-treated or targocil treated cells. Therefore, the sentence seems to be misleading by suggesting tunicamycin + targocil treatment shows better growth. The sentence should be reworded

Response: We agree, and we have revised the sentence as following (line 513-516): “In contrast, the addition of tunicamycin (0.5 μ g/ml) allowed targocil-treated *S. aureus* cells to express *vraX-lux* similarly to those treated with tunicamycin alone (Fig. 6D), indicating that accumulation of lipid-linked WTA precursors may have a positive role in the activation of VraRS.”

Lines 473-477: this sentence is also confusing for the reader. Based on Fig. S6H, targocil does have an effect on *vraX* induction, but the effect is not necessarily prolonged, in the case of Tar-3, the maximum induction point is simply time shifted and for Tar-12, there is prolonged induction, but it is overall low and never reaches the strength of induction that is seen with the other two targocil concentrations or when targocil is not present. They should rephrase the sentence.

Response: We have rephrased the sentence as following (line 516-519):” Supporting this notion, overexpression of *tarO* was able to significantly increase the expression levels of *vraX-lux* in the USA300 LAC, regardless of the presence of targocil (Fig. S9 D and E).” in the revised manuscript.

Line 561: The authors state that they showed that accumulation of flipped lipid II is a trigger for VraRS activation. As far as I can tell, the authors never directly show lipid II accumulation. They presumably make the statement based on the observed activity of the different compounds that block different steps in PG and WTA biosynthesis. They need to be more clear in the text.

Response: We agree, we have revised the sentence as following (line 602-604): “Using chemical genetics and the methods of gene overexpression, we provided evidence that the accumulation of flipped lipid II might be a trigger for the activation

of VraRS” in the revised manuscript.

Line 576: CWSS (cell wall stress stimulon?) was not introduced as abbreviation

Response: We apologize for the inconvenience, and we have corrected the “CWSS” into “cell wall stress stimulon (CWSS)”.

Lines 598 – 600: The authors never directly show accumulation of lipid-linked PG or WTA intermediates. They should rephrase the sentence.

Response: We agree. By following your comment, we deleted the sentence.

References

1. Xie Y, *et al.* (2019) *Pseudomonas savastanoi* Two-Component System RhpRS Switches between Virulence and Metabolism by Tuning Phosphorylation State and Sensing Nutritional Conditions. *mBio* 10(2).
2. Smith H (2000) Questions about the behaviour of bacterial pathogens in vivo. *Philos Trans R Soc Lond B Biol Sci* 355(1397):551-564.
3. Queck SY, *et al.* (2008) RNAIII-independent target gene control by the agr quorum-sensing system: insight into the evolution of virulence regulation in *Staphylococcus aureus*. *Mol Cell* 32(1):150-158.
4. Novick RP & Geisinger E (2008) Quorum sensing in staphylococci. *Annu Rev Genet* 42:541-564.
5. Noel A, Attwood M, Bowker K, & MacGowan A (2020) The pharmacodynamics of fosfomycin against *Staphylococcus aureus* studied in an in vitro model of infection. *Int J Antimicrob Agents* 56(1):105985.
6. Flamm RK, *et al.* (2019) Activity of fosfomycin when tested against US contemporary bacterial isolates. *Diagn Microbiol Infect Dis* 93(2):143-146.
7. Dengler V, Meier PS, Heusser R, Berger-Bachi B, & McCallum N (2011) Induction kinetics of the *Staphylococcus aureus* cell wall stress stimulon in response to different cell wall active antibiotics. *BMC Microbiol* 11:16.
8. Willing S, Schneewind O, & Missiakas D (2021) Regulated cleavage of glycan strands by the murein hydrolase SagB in *S. aureus* involves a direct interaction with LyrA (SpdC). *J Bacteriol* 203(9):e00014-00021. .
9. Bernal P, *et al.* (2010) Insertion of epicatechin gallate into the cytoplasmic membrane of methicillin-resistant *Staphylococcus aureus* disrupts penicillin-binding protein (PBP) 2a-mediated beta-lactam resistance by delocalizing PBP2. *J Biol Chem* 285(31):24055-24065.

REVIEWERS' COMMENTS

Reviewer #1 (Remarks to the Author):

My comments have been thoughtfully addressed and I enjoyed reading the revised manuscript.

Reviewer #2 (Remarks to the Author):

The authors have addressed my previous comments and strengthened the rigor of their approaches.

Reviewer #3 (Remarks to the Author):

The authors have done an excellent job of addressing most of the comments.

Only 2 very minor points need to be addressed.

For the first one, there was figure reference that does not make sense in the context of the sentence. To fix this the authors added a second figure (panel) but it still makes no sense. Page 12 of the rebuttal, to fix line 175-178. Neither Figure S2A or S2B fit into the sentence.

Second, they mislabeled a figure, also relates to page 12 of rebuttal lines 218-220. Fig. S2h is incorrectly labeled as detlatarO vs USA300 (3h). This part needs to be removed. The rest of Fig S2H seems to be correct.

Point-by-point responses to Referees' comments:

REVIEWERS' COMMENTS

Reviewer #1 (Remarks to the Author):

My comments have been thoughtfully addressed and I enjoyed reading the revised manuscript.

Response: We thank Reviewer #1 for the affirmation of our extensive revisions that addressed concerns in the previous round of revision, as well as the positive comment on our revised manuscript.

Reviewer #2 (Remarks to the Author):

The authors have addressed my previous comments and strengthened the rigor of their approaches.

Response: We appreciate the efforts of Reviewer #2 in evaluating our manuscript.

Reviewer #3 (Remarks to the Author):

The authors have done an excellent job of addressing most of the comments.

Response: Thank you. We greatly appreciated positive feedback from Reviewer #3.

Only 2 very minor points need to be addressed.

For the first one, there was figure reference that does not make sense in the context of the sentence. To fix this the authors added a second figure (panel) but it still makes no sense. Page 12 of the rebuttal, to fix line 175-178. Neither Figure S2A or S2B fit into the sentence.

Response: We sincerely thank the reviewer for careful reading and we are sorry for the inconvenience.

In this revised manuscript, we deleted the "(Fig. S2 A and B)" from the sentence and referenced the "(Fig. S2 A and B)" as following: "To validate the results of RNA-Seq, we cultured WT USA300 LAC, $\Delta tarO$ mutant, and its complementation strain ($\Delta tarO/p-tarO$) in TSB medium (Fig. S2 a and b) and carried out.... "(Line 228).

Second, they mislabeled a figure, also relates to page 12 of rebuttal lines 218-220. Fig. S2h is incorrectly labeled as detlatarO vs USA300 (3h). This part needs to be removed. The rest of Fig S2H seems to be correct.

Response: We deleted the label, that is, $\Delta tarO$ vs USA300 (3 h), from the Fig. S2H.